# CONDITIONAL POSITIONAL ENCODINGS FOR VISION TRANSFORMERS

**Xiangxiang Chu**[1], **Zhi Tian**[1], **Bo Zhang**[1], **Xinlong Wang**[2], **Chunhua Shen**[3*]
[1] Meituan Inc.   [2] Beijing Academy of AI   [3] Zhejiang University, China
{chuxiangxiang, tianzhi02, zhangbo97}@meituan.com,
xinlong.wang96@gmail.com, chunhua@me.com

## ABSTRACT

We propose a conditional positional encoding (CPE) scheme for vision Transformers (Dosovitskiy et al., 2021; Touvron et al., 2020). Unlike previous fixed or learnable positional encodings that are predefined and independent of input tokens, CPE is dynamically generated and conditioned on the local neighborhood of the input tokens. As a result, CPE can easily generalize to the input sequences that are longer than what the model has ever seen during the training. Besides, CPE can keep the desired translation equivalence in vision tasks, resulting in improved performance. We implement CPE with a simple Position Encoding Generator (PEG) to get seamlessly incorporated into the current Transformer framework. Built on PEG, we present Conditional Position encoding Vision Transformer (CPVT). We demonstrate that CPVT has visually similar attention maps compared to those with learned positional encodings and delivers outperforming results. Our Code is available at: `https://git.io/CPVT`.

## 1 INTRODUCTION

Recently, Transformers (Vaswani et al., 2017) have been viewed as a strong alternative to Convolutional Neural Networks (CNNs) in visual recognition tasks such as classification (Dosovitskiy et al., 2021) and detection (Carion et al., 2020; Zhu et al., 2021). Unlike the convolution operation in CNNs, which has a limited receptive field, the self-attention mechanism in the Transformers can capture the long-distance information and dynamically adapt the receptive field according to the image content. Consequently, Transformers are considered more flexible and powerful than CNNs, being promising to achieve more progress in visual recognition.

However, the self-attention operation in Transformers is permutation-invariant, which discards the order of the tokens in an input sequence. To mitigate this issue, previous works (Vaswani et al., 2017; Dosovitskiy et al., 2021) add the absolute positional encodings to each input token (see Figure 1a), which enables order-awareness. The positional encoding can either be learnable or fixed with sinusoidal functions of different frequencies. Despite being effective, these positional encodings seriously harm the flexibility of the Transformers, hampering their broader applications. Taking the learnable version as an example, the encodings are often a vector of equal length to the input sequence, which are jointly updated with the network weights during training. As a result, the length and the value of the positional encodings are fixed once trained. During testing, it causes difficulties of handling the sequences longer than the ones in the training data.

The inability to adapt to longer input sequences during testing greatly limits the range of generalization. For instance, in vision tasks like object detection, we expect the model can be applied to the images of any size during inference, which might be much larger than the training images. A possible remedy is to use bicubic interpolation to upsample the positional encodings to the target length, but it degrades the performance without fine-tuning as later shown in our experiments. For vision in general, we expect that the models be translation-equivariant. For example, the output feature maps of CNNs shift accordingly as the target objects are moved in the input images. However, the absolute positional encoding scheme might break the translation equivalence because it adds unique

---

*Corresponding author.

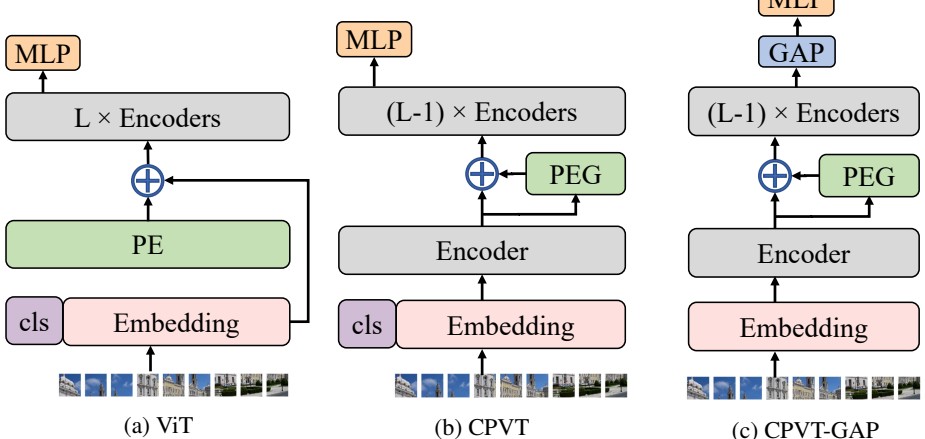

Figure 1. Vision Transformers: (a) ViT (Dosovitskiy et al., 2021) with explicit 1D learnable positional encodings (`PE`) (b) CPVT with conditional positional encoding from the proposed Position Encoding Generator (`PEG`) plugin, which is the *default* choice. (c) CPVT-GAP without class token (`cls`), but with global average pooling (GAP) over all items in the sequence. Note that GAP is a bonus version which has boosted performance.

positional encodings to each token (or each image patch). One may overcome the issue with relative positional encodings as in (Shaw et al., 2018). However, relative positional encodings not only come with extra computational costs, but also require modifying the implementation of the standard Transformers. Last but not least, the relative positional encodings cannot work equally well as the absolute ones, because the image recognition task still requires absolute position information (Islam et al., 2020), which the relative positional encodings fail to provide.

In this work, we advocate a novel positional encoding (PE) scheme to incorporate the position information into Transformers. Unlike the predefined and input-agnostic positional encodings used in previous works (Dosovitskiy et al., 2021; Vaswani et al., 2017; Shaw et al., 2018), the proposed PE is dynamically generated and conditioned on the local neighborhood of input tokens. Thus, our positional encodings can change along with the input size and try to keep translation equivalence. We demonstrate that the vision transformers (Dosovitskiy et al., 2021; Touvron et al., 2020) with our new PE (*i.e.* CPVT, see Figure 1c) achieve even better performance. We summarize our contributions as,

- We propose a novel positional encoding (PE) scheme, termed *conditional position encodings* (CPE). CPE is dynamically generated with Positional Encoding Generators (PEG) and can be effortlessly implemented by the modern deep learning frameworks (Paszke et al., 2019; Abadi et al., 2016; Chen et al., 2015), requiring no changes to the current Transformer APIs. Through an in-depth analysis and thorough experimentations, we unveil that this design affords both absolute and relative encoding yet it goes above and beyond.

- As opposed to widely-used absolute positional encodings, CPE can provide a kind of stronger explicit bias towards the **translation equivalence** which is important to improve the performance of Transformers.

- Built on CPE, we propose Conditional Position encoding Vision Transformer (CPVT). It achieves better performance than previous vison transformers (Dosovitskiy et al., 2021; Touvron et al., 2020).

- CPE can well generalize to arbitrary input resolutions, which are required in many important downstream tasks such as segmentation and detection. Through experiments we show that CPE can boost the segmentation and detection performance for pyramid transformers like (Wang et al., 2021) by a clear margin.

## 2 RELATED WORK

Since self-attention itself is permutation-equivariant (see A), positional encodings are commonly employed to incorporate the order of sequences (Vaswani et al., 2017). The positional encodings can either be fixed or learnable, while either being absolute or relative. Vision transformers follow the same fashion to imbue the network with positional information.

**Absolute Positional Encoding.** The absolute positional encoding is the most widely used. In the original transformer (Vaswani et al., 2017), the encodings are generated with the sinusoidal functions of different frequencies and then they are added to the inputs. Alternatively, the positional encodings can be learnable, where they are implemented with a fixed-dimension matrix/tensor and jointly updated with the model's parameters with SGD.

**Relative Positional Encoding.** The relative position encoding (Shaw et al., 2018) considers distances between the tokens in the input sequence. Compared to the absolute ones, the relative positional encodings can be translation-equivariant and can naturally handle the sequences longer than the longest sequences during training (*i.e.*, being inductive). A 2-D relative position encoding is proposed for image classification in (Bello et al., 2019), showing superiority to 2D sinusoidal embeddings. The relative positional encoding is further improved in XLNet (Yang et al., 2019b) and DeBERTa (He et al., 2020), showing better performance.

**Other forms.** Complex-value embeddings (Wang et al., 2019) are an extension to model global absolute encodings and show improvement. RoFormer (Su et al., 2021) utilizes a rotary position embedding to encode both absolute and relative position information for text classification. FLOATER (Liu et al., 2020) proposes a novel continuous dynamical model to capture position encodings. It is not limited by the maximum sequence length during training, meanwhile being parameter-efficient.

**Similar designs to CPE.** Convolutions are used to model local relations in ASR and machine translation (Gulati et al., 2020; Mohamed et al., 2019; Yang et al., 2019a; Yu et al., 2018). However, they are mainly limited to 1D signals. We instead process 2D vision images.

## 3 VISION TRANSFORMER WITH CONDITIONAL POSITION ENCODINGS

### 3.1 MOTIVATION

In vision transformers, an input image of size $H \times W$ is split into patches with size $S \times S$, the number of patches is $N = \frac{HW}{S^2}$[1]. The patches are added with the same number of learnable absolute positional encoding vectors. In this work, we argue that the positional encodings used here have two issues. First, it prevents the model from handling the sequences longer than the learnable PE. Second, it makes the model not translation-equivariant because a unique positional encoding vector is added to every one patch. The translation equivalence plays an important role in classification because we hope the networks' responses changes accordingly as the object moves in the image.

One may note that the first issue can be remedied by removing the positional encodings since except for the positional encodings, all other components (*e.g.*, MHSA and FFN) of the vision transformer can directly be applied to longer sequences. However, this solution severely deteriorates the performance. This is understandable because the order of the input sequence is an important clue and the model has no way to extract the order without the positional encodings. The experiment results on ImageNet are shown in Table 1. By removing the positional encodings, DeiT-tiny's performance on ImageNet dramatically degrades from 72.2% to 68.2%.

Second, in DeiT (Touvron et al., 2020), they show that we can interpolate the position encodings to make them have the same length of the longer sequences. However, this method requires fine-tuning the model a few more epochs, otherwise the performance will remarkably drop, as shown in Table 1. This goes contrary to what we would expect. With the higher-resolution inputs, we often expect a remarkable performance improvement without any fine-tuning. Finally, the relative position encodings (Shaw et al., 2018; Bello et al., 2019) can cope with both the aforementioned

---

[1]$H$ and $W$ shall be divisible by $S$, respectively.

Table 1. Comparison of various positional encoding (PE) strategies tested on ImageNet validation set in terms of the top-1 accuracy. Removing the positional encodings greatly damages the performance. The relative positional encodings have inferior performance to the absolute ones

| Model | Encoding | Top-1@224(%) | Top-1@384(%) |
|---|---|---|---|
| DeiT-tiny (Touvron et al., 2020) | ✗ | 68.2 | 68.6 |
| DeiT-tiny (Touvron et al., 2020) | learnable | 72.2 | 71.2 |
| DeiT-tiny (Touvron et al., 2020) | sin-cos | 72.3 | 70.8 |
| DeiT-tiny | 2D RPE (Shaw et al., 2018) | 70.5 | 69.8 |

issues. However, the relative positional encoding cannot provide *absolute position information*, which is also important to the classification performance (Islam et al., 2020). As shown in Table 1, the model with relative position encodings has inferior performance (70.5% vs. 72.2%).

### 3.2 CONDITIONAL POSITIONAL ENCODINGS

We argue that a successful positional encoding for vision tasks should meet these requirements,

(1) Making the input sequence *permutation-variant* and providing stronger explicit bias towards *translation-equivariance*.

(2) Being inductive and able to handle the sequences longer than the ones during training.

(3) Having the ability to provide the absolute position to a certain degree. This is important to the performance as shown in (Islam et al., 2020).

In this work, we find that characterizing the local relationship by positional encodings is sufficient to meet all of the above. First, it is *permutation-variant* because the permutation of input sequences also affects the order in some local neighborhoods. However, translation of an object in an input image does not change the order in its local neighborhood, *i.e.*, *translation-equivariant* (see Section A). Second, the model can easily generalize to longer sequences since only the local neighborhoods of a token are involved. Besides, if the absolute position of any input token is known, the absolute position of all the other tokens can be inferred by the mutual relation between input tokens. We will show that the tokens on the borders can be aware of their absolute positions due to the commonly-used zero paddings.

Therefore, we propose *positional encoding generators* (PEG) to dynamically produce the positional encodings conditioned on the local neighborhood of an input token.

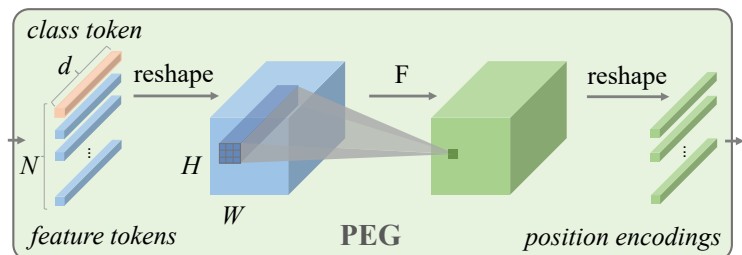

Figure 2. Schematic illustration of Positional Encoding Generator (PEG). Note $d$ is the embedding size, $N$ is the number of tokens.

**Positional Encoding Generator.** PEG is illustrated in Figure 2. To condition on the local neighbors, we first reshape the flattened input sequence $X \in \mathbb{R}^{B \times N \times C}$ of DeiT back to $X' \in \mathbb{R}^{B \times H \times W \times C}$ in the 2-D image space. Then, a function (denoted by $\mathcal{F}$ in Figure 2) is repeatedly applied to the local patch in $X'$ to produce the conditional positional encodings $E^{B \times H \times W \times C}$. PEG can be efficiently implemented with a 2-D convolution with kernel $k$ ($k \geq 3$) and $\frac{k-1}{2}$ zero paddings. Note that the zero paddings here are important to make the model be aware of the absolute positions, and $\mathcal{F}$ can be of various forms such as various types of convolutions and many others.

### 3.3 CONDITIONAL POSITIONAL ENCODING VISION TRANSFORMERS

Built on the conditional positional encodings, we propose our Conditional Positional Encoding Vision Transformers (CPVT). Except that our positional encodings are conditional, we exactly follow

ViT and DeiT to design our vision transformers and we also have three sizes CPVT-Ti, CPVT-S and CPVT-B. Similar to the original positional encodings in DeiT, the conditional positional encodings are also added to the input sequence, as shown in Figure 1 (b). In CPVT, the position where PEG is applied is also important to the performance, which will be studied in the experiments.

In addition, both DeiT and ViT utilize an extra learnable class token to perform classification (*i.e.*, cls_token shown in Figure 1 (a) and (b)). By design, the class token is not translation-invariant, although it can learn to be so. A simple alternative is to directly replace it with a global average pooling (GAP), which is inherently translation-invariant, resulting in our CVPT-GAP. Together with CPE, CVPT-GAP achieves much better image classification performance.

## 4 EXPERIMENTS

### 4.1 SETUP

**Datasets.** Following DeiT (Touvron et al., 2020), we use ILSVRC-2012 ImageNet dataset (Deng et al., 2009) with 1K classes and 1.3M images to train all our models. We report the results on the validation set with 50K images. Unlike ViT (Dosovitskiy et al., 2021), we do not use the much larger undisclosed JFT-300M dataset (Sun et al., 2017).

**Model variants.** We have three models with various sizes to adapt to various computing scenarios. The detailed settings are shown in Table 9 (see B.1). All experiments in this paper are performed on Tesla V100 machines. Training the tiny model for 300 epochs takes about 1.3 days on a single node with 8 V100 GPU cards. CPVT-S and CPVT-B take about 1.6 and 2.5 days, respectively.

**Training details** All the models (except for CPVT-B) are trained for 300 epochs with a global batch size of 2048 on Tesla V100 machines using AdamW optimizer (Loshchilov & Hutter, 2019). We do not tune the hyper-parameters and strictly comply with the settings in DeiT (Touvron et al., 2020). The learning rate is scaled with this formula $lr_{scale} = {}^{0.0005 \cdot \text{BatchSize}_{global}}/512$. The detailed hyperparameters are in the B.2.

### 4.2 GENERALIZATION TO HIGHER RESOLUTIONS

As mentioned before, our proposed PEG can directly generalize to larger image sizes without any fine-tuning. We confirm this here by evaluating the models trained with $224 \times 224$ images on the $384 \times 384, 448 \times 448, 512 \times 512$ images, respectively. The results are shown in Table 2. With the $384 \times 384$ input images, the DeiT-tiny with learnable positional encodings degrades from 72.2% to 71.2%. When equipped with sine encoding, the tiny model degrades from 72.2% to 70.8%. In constrat, our CPVT model with the proposed PEG can directly process the larger input images, and CPVT-Ti's performance is boosted from 73.4% to 74.2% when applied to $384 \times 384$ images. Our CPVT-Ti outperforms DeiT-tiny by 3.0%. This gap continues to increase as the input resolution enlarges.

Table 2. Direct evaluation on other resolutions without fine-tuning. The models are trained on $224 \times 224$. A simple PEG of a single layer of $3 \times 3$ depth-wise convolution is used here

| Model | Params | 160(%) | 224(%) | 384(%) | 448(%) | 512(%) |
|---|---|---|---|---|---|---|
| DeiT-tiny | 6M | 65.6 | 72.2 | 71.2 | 68.8 | 65.9 |
| DeiT-tiny (sin) | 6M | 65.2 | 72.3 | 70.8 | 68.2 | 65.1 |
| DeiT-tiny (no pos) | 6M | 62.1 | 68.2 | 68.6 | 68.4 | 65.0 |
| CPVT-Ti | 6M | 66.8(+1.2) | 72.4(+0.2) | 73.2(+2.0) | 71.8(+3.0) | 70.3(+4.4) |
| CPVT-Ti ‡ | 6M | 67.7 (+2.1) | 73.4(+1.2) | 74.2(+3.0) | 72.6(+3.8) | 70.8(+4.9) |
| DeiT-small | 22M | 75.6 | 79.9 | 78.1 | 75.9 | 72.6 |
| CPVT-S | 22M | 76.1(+0.5) | 79.9 | 80.4(+1.5) | 78.6(+2.7) | 76.8(+4.2) |
| DeiT-base | 86M | 79.1 | 81.8 | 79.7 | 79.8 | 78.2 |
| CPVT-B | 86M | 80.5(+1.4) | 81.9(+0.1) | 82.3(+2.6) | 82.4(+2.6) | 81.0(+2.8) |

‡: Insert one PEG each after the first encoder till the fifth encoder

### 4.3 CPVT WITH GLOBAL AVERAGE POOLING

By design, the proposed PEG is translation-equivariant (ignore paddings). Thus, if we further use the translation-invariant global average pooling (GAP) instead of the `cls_token` before the final classification layer of CPVT. CPVT can be translation-invariant, which should be beneficial to the ImageNet classification task. Note the using GAP here results in even less computation complexity because we do not need to compute the attention interaction between the class token and the image patches. As shown in Table 3, using GAP here can boost CPVT by more than **1%**. For example, equipping CPVT-Ti with GAP obtains 74.9% top-1 accuracy on the ImageNet validation dataset, which outperforms DeiT-tiny by a large margin (+2.7%). Moreover, it even exceeds DeiT-tiny model with distillation (74.5%). In contrast, DeiT with GAP cannot gain so much improvement (only 0.4% as shown in Table 3) because the original learnable absolute PE is not translation-equivariant and thus GAP with the PE is not translation-invariant. Given the superior performance, we hope our model can be a strong PE alternative in vision transformers.

Table 3. Performance comparison of Class Token (CLT) and global average pooling (GAP) on ImageNet. CPVT's can be further boosted with GAP

| Model | Head | Params | Top-1 Acc (%) | Top-5 Acc (%) |
|---|---|---|---|---|
| DeiT-tiny (Touvron et al., 2020) | CLT | 6M | 72.2 | 91.0 |
| DeiT-tiny | GAP | 6M | 72.6 | 91.2 |
| CPVT-Ti ‡ | CLT | 6M | 73.4 | 91.8 |
| CPVT-Ti ‡ | GAP | 6M | **74.9** | **92.6** |
| DeiT-small (Touvron et al., 2020) | CLT | 22M | 79.9 | 95.0 |
| DeiT-small | GAP | 22M | 80.2 | 95.2 |
| CPVT-S ‡ | CLT | 23M | 80.5 | 95.2 |
| CPVT-S ‡ | GAP | 23M | **81.5** | **95.7** |

‡: Insert one PEG each after the first encoder till the fifth encoder

### 4.4 COMPLEXITY OF PEG

**Few Parameters.** Given the model dimension $d$, the extra number of parameters introduced by PEG is $d \times l \times k^2$ if we choose $l$ depth-wise convolutions with kernel $k$. If we use $l$ separable convolutions, this value becomes $l(d^2 + k^2 d)$. When $k = 3$ and $l = 1$, CPVT-Ti ($d = 192$) brings about 1, 728 parameters. Note that DeiT-tiny utilizes learnable position encodings with $192 \times 14 \times 14 = 37632$ parameters. Therefore, CPVT-Ti has 35, 904 fewer number of parameters than DeiT-tiny. Even using 4 layers of separable convolutions, CPVT-Ti introduces only $38952 - 37632 = 960$ more parameters, which is negelectable compared to the 5.7M model parameters of DeiT-tiny.

**FLOPs.** As for FLOPs, $l$ layers of $k \times k$ depth-wise convolutions possesses $14 \times 14 \times d \times l \times k^2$ FLOPS. Taking the tiny model for example, it involves $196 \times 192 \times 9 = 0.34M$ FLOPS for the simple case $k = 3$ and $l = 1$, which is neglectable because the model has 2.1G FLOPs in total.

### 4.5 PERFORMANCE COMPARISON

We evaluate the performance of CPVT models on the ImageNet validation dataset and report the results in Table 4. Compared with DeiT, *CPVT models have much better top-1 accuracy with similar throughputs.* Our models can enjoy performance improvement when inputs are upscaled without fine-tuning, while DeiT degrades as discussed in Table 2, see also Figure 3 for a clear comparison. Noticeably, Our model with GAP marked a new state-of-the-art for vision Transformers.

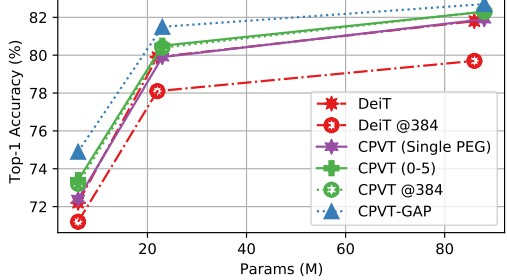

Figure 3. Comparison of CPVT and DeiT models under various configurations. Note CPVT@384 has improved performance. More PEGs can result in better performance. CPVT-GAP is the best.

Table 4. Comparison with ConvNets and Transformers on ImageNet and ImageNet Real (Beyer et al., 2020). CPVT have much better performance compared with prior Transformers

| Models | Params(M) | Input | throughput* | ImNet top-1 % | Real top-1 % |
|---|---|---|---|---|---|
| ResNet-50 (He et al., 2016) | 25 | $224^2$ | 1226.1 | 76.2 | 82.5 |
| ResNet-101 (He et al., 2016) | 45 | $224^2$ | 753.6 | 77.4 | 83.7 |
| ResNet-152 (He et al., 2016) | 60 | $224^2$ | 526.4 | 78.3 | 84.1 |
| RegNetY-4GF (Radosavovic et al., 2020) | 21 | $224^2$ | 1156.7 | 80.0 | 86.4 |
| EfficientNet-B0 (Tan & Le, 2019) | 5 | $224^2$ | 2694.3 | 77.1 | 83.5 |
| EfficientNet-B1 (Tan & Le, 2019) | 8 | $240^2$ | 1662.5 | 79.1 | 84.9 |
| EfficientNet-B2 (Tan & Le, 2019) | 9 | $260^2$ | 1255.7 | 80.1 | 85.9 |
| EfficientNet-B3 (Tan & Le, 2019) | 12 | $300^2$ | 732.1 | 81.6 | 86.8 |
| EfficientNet-B4 (Tan & Le, 2019) | 19 | $380^2$ | 349.4 | 82.9 | 88.0 |
| ViT-B/16 (Dosovitskiy et al., 2021) | 86 | $384^2$ | 85.9 | 77.9 | - |
| ViT-L/16 | 307 | $384^2$ | 27.3 | 76.5 | - |
| DeiT-tiny w/o PE (Touvron et al., 2020) | 6 | $224^2$ | 2536.5 | 68.2 | - |
| DeiT-tiny (Touvron et al., 2020) | 6 | $224^2$ | 2536.5 | 72.2 | 80.1 |
| DeiT-tiny (sine) | 6 | $224^2$ | 2536.5 | 72.3 | 80.3 |
| CPVT-Ti ‡ | 6 | $224^2$ | 2500.7 | 73.4 | 81.3 |
| **CPVT-Ti-GAP**‡ | 6 | $224^2$ | 2520.1 | **74.9** | 82.5 |
| DeiT-tiny (Touvron et al., 2020)⚗ | 6 | $224^2$ | 2536.5 | 74.5 | 82.1 |
| **CPVT-Ti**⚗ | 6 | $224^2$ | 2500.7 | **75.9** | 83.0 |
| DeiT-small (Touvron et al., 2020) | 22 | $224^2$ | 940.4 | 79.9 | 85.7 |
| CPVT-S ‡ | 23 | $224^2$ | 930.5 | 80.5 | 86.0 |
| **CPVT-S-GAP**‡ | 23 | $224^2$ | 942.3 | **81.5** | 86.6 |
| DeiT-base (Touvron et al., 2020) | 86 | $224^2$ | 292.3 | 81.8 | 86.7 |
| CPVT-B ‡ | 88 | $224^2$ | 285.5 | 82.3 | 87.0 |
| **CPVT-B-GAP**‡ | 88 | $224^2$ | 290.2 | **82.7** | 87.7 |

    *: Measured in *img/s* on a 16GB V100 GPU as in (Touvron et al., 2020).
    ‡: Insert one PEG each after the first encoder till the fifth encoder
    ⚗: trained with hard distillation using RegNetY-160 as the teacher.

We further train CPVT-Ti and DeiT-tiny using the aforementioned training settings plus the hard distillation proposed in (Touvron et al., 2020). Specifically, we use RegNetY-160 (Radosavovic et al., 2020) as the teacher. CPVT obtains 75.9%, exceeding DeiT-tiny by 1.4%.

## 4.6 PEG on Pyramid Transformer Architectures

PVT (Wang et al., 2021) is a vision transformer with the multi-stage design like ResNet (He et al., 2016). Swin (Liu et al., 2021) is a follow-up work and comes with higher performance. We apply our method on both to demonstrate its generalization ability.

**ImageNet classification.** Specifically, we remove its learnable PE and apply our PEG in position 0 of each stage with a GAP head. We use the same training settings to make a fair comparison and show the results in Table 13. Our method can significantly boost PVT-tiny by 3.1% and Swin-tiny by 1.15% on ImageNet (c.f. B.5). We also evaluate the performance of PEG on some downstream semantic segmentation and object detection tasks (see B.6). Note these tasks usually handle the various input resolutions as the training because multi-scale data augmentation is extensively used.

## 5 Ablation Study

### 5.1 Positional encoding or merely a hybrid?

One might suspect that the PEG's improvement comes from the extra *learnable parameters* introduced by the convolutional layers in PEG, instead of the local relationship retained by PEG. One

way to test the function of PEG is only adding it when calculating Q and K in the attention layer, so that only the positional information of PEG is passed through. We can achieve 71.3% top-1 accuracy on ImageNet with DeiT-tiny. This is significantly better than DeiT-tiny w/o PE (68.2%) and is similar to the one with PEG on Q, K and V (72.4%), which suggests that PEG mainly serves as a positional encoding scheme.

We also design another experiment to remove this concern. By randomly-initializing a 3×3 PEG and fixing its weights during the training, we can obtain 71.3% accuracy (Table 5), which is much higher (3.1%↑) than DeiT without any PE (68.2%). Since the weights of PEG are fixed and the performance improvement can only be due to the introduced position information. On the contrary, when we exhaustively use 12 convolutional layers (kernel size being 1, *i.e.*, not producing local relationship) to replace the PEG, these layers have much more learnable parameters than PEG. However, it only boosts the performance by 0.4% to 68.6%.

Table 5. Positional encoding rather than added parameters gives the most improvement

| Kernel | Style | Params (M) | Top-1 Acc (%) |
|---|---|---|---|
| none | - | 5.68 | 68.2 |
| 3 | fixed (random init) | 5.68 | 71.3 |
| 3 | fixed (learned init) | 5.68 | 72.3 |
| 1 (12 ×) | learnable | 6.13 | 68.6 |
| 3 | learnable | 5.68 | **72.4** |

Another interesting finding is that fixing a learned PEG also helps training. When we initialize with a learned PEG instead of the random values and train the tiny version of the model from scratch while keeping the PEG fixed, the model can also achieve 72.3% top-1 accuracy on ImageNet. This is very close to the learnable PEG (72.4%).

## 5.2 PEG POSITION IN CPVT

We also experiment by varying the position of the PEG in the model. Table 6 (left) presents the ablations for variable positions (denoted as PosIdx) based on the tiny model. *We consider the input of the first encoder by index -1.* Therefore, position 0 is the output of the first encoder block. PEG shows strong performance (∼72.4%) when it is placed at [0, 3].

Note that positioning the PEG at 0 can have much better performance than positioning it at -1 (*i.e.*, before the first encoder), as shown in Table 6 (left). We observe that the difference between the two situations is they have different receptive fields. Specifically, the former has a global field while the latter can only see a local area. Hence, *they are supposed to work similarly well if we enlarge the convolution's kernel size*. To verify our hypothesis, we use a quite large kernel size 27 with a padding size 13 at position -1, whose result is reported in Table 6 (right). It achieves similar performance to the one positioning the PEG at 0 (72.5%), which verifies our assumption.

Table 6. Comparison of different plugin positions (left) and kernels (right) using DeiT-tiny

| PosIdx | Top-1 (%) | Top-5 (%) |
|---|---|---|
| none | 68.2 | 88.7 |
| −1 | 70.6 | 90.2 |
| 0 | **72.4** | **91.2** |
| 3 | 72.3 | 91.1 |
| 6 | 71.7 | 90.8 |
| 10 | 69.0 | 89.1 |

| PosIdx | kernel | Params | Top-1 (%) | Top-5 (%) |
|---|---|---|---|---|
| -1 | 3×3 | 5.7M | 70.6 | 90.2 |
| -1 | 27×27 | 5.8M | **72.5** | **91.3** |

## 5.3 COMPARISONS WITH OTHER POSITIONAL ENCODINGS

We compare PEG with other commonly used encodings: absolute positional encoding (e.g. sinusoidal (Vaswani et al., 2017)), *relative positional encoding* (RPE) (Shaw et al., 2018) and *learnable encoding* (LE) (Devlin et al., 2019; Radford et al., 2018), as shown in Table 7.

DeiT-tiny obtains 72.2% with the learnable absolute PE. We experiment with the 2-D sinusoidal encodings and it achieves on-par performance. For RPE, we follow (Shaw et al., 2018) and set the

Table 7. Comparison of various positional encoding strategies. LE: learnable positional encoding. RPE: relative positional encoding

| Model | PEG Pos | Encoding | Top-1 (%) | Top-5 (%) |
|---|---|---|---|---|
| DeiT-tiny (2020) | - | LE | 72.2 | 91.0 |
| DeiT-tiny | - | 2D sin-cos | 72.3 | 91.0 |
| DeiT-tiny | - | 2D RPE | 70.5 | 90.0 |
| CPVT-Ti | 0-1 | PEG | 72.4 | 91.2 |
| CPVT-Ti | 0-1 | PEG + LE | 72.9 | 91.4 |
| CPVT-Ti | 0-1 | 4×PEG + LE | 72.9 | 91.4 |
| **CPVT-Ti** | 0-5 | PEG | **73.4** | **91.8** |

local range hyper-parameter $K$ as 8, with which we obtain 70.5%. RPE here does not encode any absolute position information, see discussion in D.1 and B.3.

Moreover, we combine the learnable absolute PE with a single-layer PEG. This boosts the baseline CPVT-Ti (0-1) by 0.5%. If we use 4-layer PEG, it can achieve 72.9%. If we add a PEG to each of the first five blocks, we can obtain 73.4%, which is better than stacking them within one block.

**CPE is not a simple combination of APE and RPE.** We further compare our method with a baseline with combination of APE and RPE. Specifically, we use learnable positional encoding (LE) as DeiT at the beginning of the model and supply 2D RPE for every transformer block. This setting achieves 72.4% top-1 accuracy on ImageNet, which is comparable to a single PEG (72.4%). Nevertheless, this experiment does not necessarily indicate that our CPE is a simple combination of APE and RPE. When tested on different resolutions, this baseline cannot scale well compared to ours (Table 8). RPE is not able to adequately mitigate the performance degradation on top of LE. This shall be seen as a major difference.

Table 8. Direct evaluation on other resolutions without fine-tuning. The models are trained on 224×224. CPE outperforms LE+RPE combination on untrained resolutions.

| Model | Positional Params | 160(%) | 224(%) | 384(%) | 448(%) | 512(%) |
|---|---|---|---|---|---|---|
| DeiT-tiny (LE+RPE) | 40011 | 65.6 | 72.4 | 70.8 | 68.4 | 65.6 |
| DeiT-tiny (PEG at Pos 0) | 1920 | 66.8 | 72.4 | 73.2 | 71.8 | 70.3 |

**PEG can continuously improve the performance if stacked more.** We use LE not only at the beginning but also in the next 5 layers to have a similar thing as 0-5 PEG configuration.This setting achieves 72.7% top-1 accuracy on ImageNet, which is 0.7% lower than PEG (0-5). This setting suggests that it is also beneficial to have more of LEs, but not as good as ours. It is expected since we exploit relative information via PEGs at the same time.

## 6    CONCLUSION

We introduced CPVT, a novel method to provide the position information in vision transformers, which dynamically generates the position encodings based on the local neighbors of each input token. Through extensive experimental studies, we demonstrate that our proposed positional encodings can achieve stronger performance than the previous positional encodings. The transformer models with our positional encodings can naturally process longer input sequences and keep the desired translation equivalence in vision tasks. Moreover, our positional encodings are easy to implement and come with negligible cost. We look forward to a broader application of our method in transformer-driven vision tasks like segmentation and video processing.

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

## A  TRANSLATION EQUIVARIANCE

The term translation-equivariance means the output feature maps can be equally translated with the input signal. Imagine there is a person in the left-top of an image, if the person is moved to the right-bottom, the output feature maps will change accordingly. This property is very important to the success of convolution network. Convolution (ignoring paddings), RPE, and self-attention are all translation-equivariant operations (regardless of their receptive field). It's nontrivial to make absolute positional encodings like DeiT (using learnable positional encoding) translation-equivariant since different absolute positions will be added if the input signal is translated. Note that our method is not strictly translation-equivariant because of the zero padding. Instead, it provides a kind of stronger explicit bias towards the translation-equivariant property.

## B  EXPERIMENT DETAILS

### B.1  ARCHITECTURE VARIANTS OF CPVT

Table 9. CPVT architecture variants. The larger model, CPVT-B, has the same architecture as ViT-B (Dosovitskiy et al., 2021) and DeiT-B (Touvron et al., 2020). CPVT-S and CPVT-Ti have the same architecture as DeiT-small and DeiT-tiny respectively

| Model | #channels | #heads | #layers | #params |
|-------|-----------|--------|---------|---------|
| CPVT-Ti | 192 | 3 | 12 | 6M |
| CPVT-S | 384 | 6 | 12 | 22M |
| CPVT-B | 768 | 12 | 12 | 86M |

### B.2  THE HYPERPARAMETERS OF CPVT

As for the ImageNet classification task, we use exactly the same hyperparameters as DeiT except for the base model because it is not always stably trained using AdamW. The detailed setting is shown in Table 10.

Table 10. Hyper-parameters for ViT, DeiT and CPVT

| Methods | ViT | DeiT | CPVT |
|---------|-----|------|------|
| Epochs | 300 | 300 | 300 |
| Batch size | 4096 | 1024 | 1024 |
| Optimizer | AdamW | AdamW | LAMB |
| Learning rate decay | cosine | cosine | cosine |
| Weight decay | 0.3 | 0.05 | 0.05 |
| Warmup epochs | 3.4 | 5 | 5 |
| Label smoothing $\varepsilon$ (Szegedy et al., 2016) | ✗ | 0.1 | 0.1 |
| Dropout (Srivastava et al., 2014) | 0.1 | ✗ | ✗ |
| Stoch. Depth (Huang et al., 2016) | ✗ | 0.1 | 0.1 |
| Repeated Aug (Hoffer et al., 2020) | ✗ | ✓ | ✓ |
| Gradient Clip. | ✓ | ✗ | ✗ |
| Rand Augment (Cubuk et al., 2020) | ✗ | 9/0.5 | 9/0.5 |
| Mixup prob. (Zhang et al., 2018) | ✗ | 0.8 | 0.8 |
| Cutmix prob. (Yun et al., 2019) | ✗ | 1.0 | 1.0 |
| Erasing prob. (Zhong et al., 2020) | ✗ | 0.25 | 0.25 |

### B.3  IMPORTANCE OF ZERO PADDINGS

We design an experiment to verify the importance of the *zero paddings*, which can help the model infer the absolute positional information. Specifically, we use CPVT-S and simply remove the zero paddings from CPVT while keeping all other settings unchanged. Table 11 shows that this can only obtain 70.5%, which indicates that the zero paddings and absolute positional information play important roles in classifying objects.

Table 11. Ablation study on ImageNet performance w/ or w/o zero paddings

| Model | Padding | Top-1 Acc(%) | Top-5 Acc(%) |
|---|---|---|---|
| CPVT-Ti | ✓ | **72.4** | **91.2** |
| | ✗ | 70.5 | 89.8 |

## B.4 Single PEG vs. Multiple PEGs

We further evaluate whether or not using *multi-position* encodings can benefit the performance in Table 12. Notice we denote by $i$-$j$ the inserted positions of PEG which start from the $i$-th encoder and end at the $j-1$-th one (inclusion). By inserting PEGs to five positions, the top-1 accuracy of the tiny model can achieve 73.4%, which surpasses DeiT-tiny by 1.2%. Similarly, CPVT-S can achieve 80.5%. It turns out more PEGs do help, but up to a level where more PEGs become incremental (0-5 vs. 0-11).

Table 12. CPVT's sensitivity to number of plugin positions

| Positions | Model | Params (M) | Top-1 Acc (%) | Top-5 Acc (%) |
|---|---|---|---|---|
| 0-1 | tiny | 5.7 | 72.4 | 91.2 |
| 0-5 | tiny | 5.9 | **73.4** | **91.8** |
| 0-11 | tiny | 6.1 | **73.4** | **91.8** |
| 0-1 | small | 22.0 | 79.9 | 95.0 |
| 0-5 | small | 22.9 | 80.5 | **95.2** |
| 0-11 | small | 23.8 | **80.6** | **95.2** |

## B.5 Classfication Evaluation of Swin with PEG

We show the validation curves when training Swin (Liu et al., 2021) equipped with PEG in Figure 4. It can boost Swin-tiny from 81.10% to 82.25% (+1.15%↑) on ImageNet.

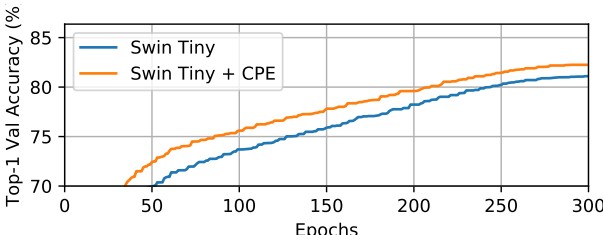

Figure 4. CPE boosts Swin Tiny on ImageNet by 1.15% top-1 Acc.

## B.6 Evaluation on Segmentation and Detection

**Semantic segmentation on ADE20K.** We evaluate the performance of PEG on the ADE20K (Zhou et al., 2017) segmentation task. Based on the Semantic FPN framework (Kirillov et al., 2019), PVT achieves much better results than ResNet (He et al., 2016) baselines. Under carefully controlled settings, PEG further boosts PVT-tiny by 3.1% mIoU.

**Object detection on COCO.** We also perform controlled experiments with the RetinaNet (Lin et al., 2017) framework on the COCO detection task. The results are shown in Table 13. In the standard 1× schedule, PEG improves PVT-tiny by 2.0% mAP. PEG brings 2.4% higher mAP under the 3 × schedule.

Table 13. Our method boosts the performance of PVT on ImageNet classification, ADE20K segmentation and COCO detection

| Backbone | ImageNet | | Semantic FPN on ADE20K | | RetinaNet on COCO | | |
|---|---|---|---|---|---|---|---|
| | Params (M) | Top-1 (%) | Params (M) | mIoU (%) | Params (M) | mAP (%, 1×) | mAP (%, 3×, +MS) |
| ResNet-18 (He et al., 2016) | 12 | 69.8 | 16 | 32.9 | 21 | 31.8 | 35.4 |
| PVT-tiny (Wang et al., 2021) | 13 | 75.0 | 17 | 35.7 | 23 | 36.7 | 39.4 |
| PVT-tiny+PEG | 13 | 77.3 | 17 | 38.0 | 23 | 38.0 | **41.8** |
| PVT-tiny+GAP | 13 | 75.9 | 17 | 36.0 | 23 | 36.9 | 39.7 |
| PVT-tiny+PEG+GAP | 13 | **78.1** | 17 | **38.8** | 23 | **38.7** | **41.8** |
| PVT-small (Wang et al., 2021) | 25 | 79.8 | 28 | 39.8 | 34 | 40.4 | 42.2 |
| PVT-small+PEG+GAP | 25 | **81.2** | 28 | **44.3** | 34 | **43.0** | **45.2** |
| PVT-Medium (Wang et al., 2021) | 44 | 81.2 | 48 | 41.6 | 54 | 41.9 | 43.2 |
| PVT-Medium+PEG+GAP | 44 | **82.7** | 48 | **44.9** | 54 | **44.3** | **46.4** |

### B.7 ABLATION ON OTHER FORMS OF PEG

We explore several forms of PEG based on the tiny model, which change the type of convolution, kernel size and layers. The inserted position is 0. The result is shown in Table 14. When we use large kernel of 7×7 or dense convolution, the performance improvement is limited. Stacking more layers of depth-wise convolution doesn't bring significant improvement. Therefore, we use the simplest form as our default implementation. It indicates that this design is enough to provide good position information.

Table 14. Other forms of PEG. The simple form of a single depth-wise 3×3 is good enough.

| Variants | Model | Top-1 Acc (%) |
|---|---|---|
| 1 Depthwise Conv 3×3 | tiny | 72.4 |
| 1 Depthwise Conv 7×7 | tiny | 72.5 |
| 4 * (Depthwise Conv 3×3 +BN+ReLU) | tiny | 72.4 |
| 1 Dense Conv 3×3 | tiny | 72.3 |
| 4 * (Dense Conv 3×3+BN+ReLU) | tiny | 72.5 |

## C EXAMPLE CODE

### C.1 PEG

In the simplest form, we use a single depth-wise convolution and show its usage in Transformer by the following PyTorch snippet. Through experiments, we find that such a simple design (*i.e.*, depth-wise 3×3) readily achieves on par or even better performance than the recent SOTAs. We give the torch implementation example in Alg. 1.

## D MORE DISCUSSIONS

### D.1 WHY RPE WORKS LESS WELL THAN ABSOLUTE PE?

As mentioned in Section 5.3 (main text), RPE is inferior to the absolute positional encoding. It is because RPE does not encode any absolute position information. Also discussed in Section B.3 (main text), absolute position information is also important even for ImageNet classification as it is needed to determine which object is at the center of the image. Note that there might be multiple objects in an image, and the label of an image is the category of the object at the center.

Additionally, although RPE becomes popular recently, it is often jointly used with absolute positional encodings (e.g., in ConViT (d'Ascoli et al., 2021)), or the absolute position information is leaked in other ways (e.g., convolution paddings in CoAtNet (Dai et al., 2021)). This further suggests absolute position information is crucial.

---

**Algorithm 1** PyTorch snippet of PEG.

---

```python
import torch
import torch.nn as nn
class VisionTransformer:
  def __init__(layers=12, dim=192, nhead=3, img_size=224, patch_size=16):
    self.pos_block = PEG(dim)
    self.blocks = nn.ModuleList([TransformerEncoderLayer(dim, nhead, dim*4) for _ in range(
        layers)])
    self.patch_embed = PatchEmbed(img_size, patch_size, dim*4)
  def forward_features(self, x):
    B, C, H, W = x.shape
    x, patch_size = self.patch_embed(x)
    _H, _W = H // patch_size, W // patch_size
    x = torch.cat((self.cls_tokens, x), dim=1)
    for i, blk in enumerate(self.blocks):
      x = blk(x)
      if i == 0:
        x = self.pos_block(x, _H, _W)
    return x[:, 0]

class PEG(nn.Module):
  def __init__(self, dim=2\textsc{56}, k=3):
    self.pos = nn.Conv2d(dim, dim, k, 1, k//2, groups=dim) # Only for demo use, more
        complicated functions are effective too.
  def forward(self, x, H, W):
    B, N, C = x.shape
    cls_token, feat_tokens = x[:, 0], x[:, 1:]
    feat_tokens = feat_tokens.transpose(1, 2).view(B, C, H, W)
    x = self.pos(feat_tokens) + feat_tokens
    x = x.flatten(2).transpose(1, 2)
    x = torch.cat((cls_token.unsqueeze(1), x), dim=1)
    return x
```

---

## D.2 COMPARISON TO LAMBDA NETWORKS

Our work is also related to Lambda Networks (Bello, 2021) which uses 2D relative positional encodings. We evaluate its lambda module with an embedding size of 128, where we denote its encoding scheme as RPE2D-d128. Noticeably, this configuration has about 5.9M parameters (comparable to DeiT-tiny) but only obtains 68.7%. We attribute its failure to the limited ability in capturing the correct positional information. After all, lambda layers are designed with the help of many CNN backbones components such as down-sampling to form various stages, to replace ordinary convolutions in ResNet (He et al., 2016). In contrast, CPVT is transformer-based.

## D.3 QUALITATIVE ANALYSIS OF CPVT

Thus far, we have shown that PEG can have better performance than the original positional encodings. However, because PEG provides the position in an implicit way, it is interesting to see if PEG can indeed provide the position information as the original positional encodings. Here we investigate this by visualizing the attention weights of the transformers. Specifically, given a 224×224 image (i.e. 14×14 patches), the score matrix within a single head is 196×196. We visualize the normalized self-attention score matrix of the second encoder block.

We first visualize the attention weights of DeiT with the original positional encodings. As shown in Figure 5 (middle), the diagonal element interacts strongly with its local neighbors but weakly with those far-away elements, which suggests that DeiT with the original positional encodings learn to attend the local neighbors of each patch. After the positional encodings are removed (denoted by DeiT w/o PE), all the patches produce similar attention weights and fail to attend to the patches near themselves, see Figure 5 (left).

Finally, we show the attention weights of our CPVT model with PEG. As shown in Figure 5 (right), like the original positional encodings, the model with PEG can also learn a similar attention pattern, which indicates that the proposed PEG can provide the position information as well.

We illustrate the attention scores in several encoder blocks of DeiT (Touvron et al., 2020) and CPVT in the Fig. 6. It shows both methods learn similar locality patterns. As attention scores are computed over the tokens projected in different subspaces (Q and K), they do not necessarily show a strict diagonal pattern, where some may have slight shift, see DeiT in Fig. 6c and CPVT of Fig. 5 right.



Figure 5. Normalized attention scores (first head) of the second encoder block of DeiT without position encoding (DeiT w/o PE), DeiT (Touvron et al., 2020), and CPVT on the same input sequence. Position encodings are key to developing a schema of locality in lower layers of DeiT. Meantime, CPVT profits from conditional encodings and follows a similar locality pattern.

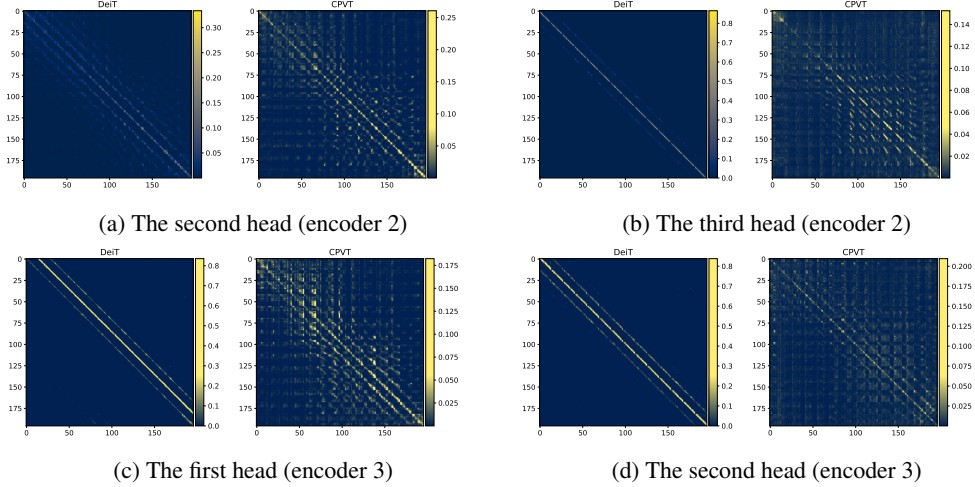

Figure 6. Normalized attention scores (the second and third head) of the second and third encoder block of DeiT (Touvron et al., 2020), and CPVT on the same input sequence. DeiT and CPVT share similar locality patterns that are aligned diagonally (some might shift).

## D.4  COMPARISON WITH OTHER APPROACHES

We further compare our method with other approaches such as CvT (Wu et al., 2021), ConViT (d'Ascoli et al., 2021) and CoAtNet (Dai et al., 2021) on ImageNet validation set in Table 15. To make fair comparisons, we categorize these methods into two groups: plain and pyramid models. Since our models are primarily for plain models, we adapt our methods on two popular pyramid frameworks PVT and Swin. Our CPVT-S-GAP slightly outperforms ConViT-S by 0.2% with 4M fewer parameters and 0.8G fewer FLOPs. When equipped with pyramid designs, our methods are still comparable to CvT and CoAtNet.

**Comparison with DeiT w/ Convolutional Projection.**  Note CvT uses a depth-wise convolution in $q$-$k$-$v$ projection which they call it *Convolutional Projection*. Instead of using it in all layers, we put only one of such design into DeiT-tiny and train such a model from scratch under strictly controlled settings. We insert it in the position 0 as in our method. The result is shown in Table 16. This CvT-flavored DeiT achieves 70.6% top-1 accuracy on ImageNet validation set, which is lower than ours (72.4%). Note that $q$-$k$-$v$ projections in CvT utilize three depthwise convolutions, therefore, this setting has more parameters than ours. This attests the difference of CvT and CPVT, verifying our advantage by learning better position encodings other than inserting them in all layers to have the ability to capture local context and to remove ambiguity in attention.

Table 15. Performance comparison with other approaches such as CvT (Wu et al., 2021), ConViT (d'Ascoli et al., 2021) and CoAtNet (Dai et al., 2021) on ImageNet validation set. All the models are trained on ImageNet-1k dataset and tested on the validation set using 224×224 resolution.

| Model | Type | Params | FLOPs | Top-1 Acc (%) |
|---|---|---|---|---|
| DeiT-small (Touvron et al., 2020) | Plain | 22M | 4.6G | 79.9 |
| ConViT-S (d'Ascoli et al., 2021) | Plain | 27M | 5.4G | 81.3 |
| CPVT-S-GAP (ours) | Plain | 23M | 4.6G | **81.5** |
| CoAtNet-0 (Dai et al., 2021) | Pyramid | 25M | 4.2G | 81.6 |
| CvT-13 (Wu et al., 2021) | Pyramid | 20M | 4.5G | 81.6 |
| PVT-small (Wang et al., 2021) | Pyramid | 25M | 3.8G | 79.8 |
| PVT-small+PEG+GAP | Pyramid | 25M | 3.8G | 81.2 |
| Swin-tiny (Liu et al., 2021) | Pyramid | 29M | 4.5G | 81.3 |
| Swin-tiny+PEG+GAP | Pyramid | 29M | 4.5G | **82.3** |

Table 16. Comparison with positional encoding in CvT (Wu et al., 2021) on ImageNet validation set. All the models are trained on ImageNet-1k dataset and tested on the validation set using 224×224 resolution.

| Model | Params | Insert Position | Top-1 Acc (%) |
|---|---|---|---|
| CPVT-Ti | 5681320 | 0 | 72.4 |
| DeiT+ Convolutional Projection | 5685352 | 0 | 70.6 |

