# OpenReview forum: "Conditional Positional Encodings for Vision Transformers"
_ICLR.cc/2023/Conference — ICLR 2023 poster_

### Official Review · Reviewer_tvtM · 2022-10-24

**Confidence:** 4
**Correctness:** 4
**Technical Novelty And Significance:** 3
**Empirical Novelty And Significance:** 3
**Recommendation:** 8

**Clarity, Quality, Novelty And Reproducibility:**

Clarity: The paper is clear.

Quality: The paper is of high quality.

Novelty: Method is fairly novel, but is somewhat related to architectures that combine convolutional and self-attention layers.

Reproducibility: The method seems reproducible from the information given in the paper. The appendix is particularly well written in this respect.

**Strength And Weaknesses:**

Strengths:

Comprehensive experiments demonstrating that PEG outperforms other positional encoding strategies under a variety of situations.

Clever ablations show that the gain is not coming from additional parameters introduced by the CPE.

Careful details of hyperparameters and architecture choices in the appendix.

Weaknesses:

One concern I have is that the majority of experiments were conducted on the "tiny" versions of DeiT/ViT/PVT/Swin. Could the benefits of CPEs only be significant in the "small model" regime? This would correspond with conventional wisdom that convolutions provide inductive biases that makes training with low data or small models easier. Additional experiments that illustrate how CPEs perform under model scaling should be conducted.

I also don't completely buy the argument that CPEs allow transformers to have longer input sequences. As shown in section 4.4, positional embeddings account for a very small proportion of parameters in a transformer, and if a longer input sequence is desired one could just add more positional embeddings at very little marginal cost. The real bottleneck is the quadratic complexity of transformers, which CVPT does not address.

In section 3.2 it is noted that zero-padding allows CPEs to know absolute position. My understanding is that this is because the convolution in the PEG learns that zeros => border. However, this only works for patches that are within kernel size of the border. How do the other patches get absolute location information?

The sentence at the end of page 8 could be clearer.


**Summary Of The Paper:**

The authors propose a simple modification to the positional encoding mechanism of vision transformers. Instead of using fixed sinusoidal embeddings or learned embeddings, the authors introduce conditional positional encodings (CPE), in which positional embeddings for a given patch are calculated as a learned function of that patch's local neighborhood. CPEs can be easily implemented as a single convolution with padding.

Comprehensive experiments using DeiT on Imagenet classification show that CPEs significantly improves accuracy and out perform alternative positional encoding methods. Qualitative visualizations show reasonable positional encoding. And various ablations show CPEs efficacy on additional architectures like Swin and PVT, and rule out the possibility that gains are coming simply from adding additional parameters to the model.

**Summary Of The Review:**

The authors propose a novel and effective method to replace absolute and learned positional embeddings. Through numerous experiments and ablations they show that this method is able to improve significantly upon existing methods. While the paper is comprehensive, I have a few questions that I would like addressed (see weaknesses section).

---

> ### Author Response · Authors · 2022-11-12
> **CPE boosts performance for longer sequences**
>
> **Q1:** CPE with larger models.
>
> **A1:** Many experiments are done based on tiny models in order to save GPU cost. It does not imply that CPEs are only useful in the "small model" regime. We also conducted experiments with the DeiT base model, and the results are shown in the last block of Table 4. DeiT-base is considered a large model, which has 88M parameters and is comparable with ResNet-152 with 60M parameters.
> Moreover, we further test our method on the PVT small and medium models and update the results in Table 12.  Compared with the improvements for tiny models, our method can improve these models more significantly on ADE20k segmentation and COCO detection.
>
> Table r3. Segmentation on AED20k.
> | Models    | ImageNet Top-1(%) |  Params(M)      |  mIoU（%）|
> |--------|:---------:|------:|----:|
> | ResNet-18 |  69.8| 16 |32.9
> | PVT-tiny|    75.0 | 17|35.7
> | PVT-tiny+PEG  | 77.3 | 17 | 38.0
> | PVT-tiny+GAP|   75.9 | 17 | 36.0
> | PVT-tiny+PEG+GAP  |78.1| 17|38.8
> |PVT-small| 79.8 |28|39.8
> |PVT-small+PEG+GAP | 81.2| 28|44.3
> |PVT-Medium| 81.2 |48|41.6
> |PVT-Medium+PEG+GAP|82.7| 48|44.9
>
> Table r4. Object Detection on COCO.
>
> | Models    |   Params(M)      |  1x mAP（%）| 3x mAP (%)|
> |--------|:---------:|------:|----:|
> | PVT-tiny|   23|36.7|39.4
> | PVT-tiny+PEG  |23|38.0| 41.8
> | PVT-tiny+GAP|  23| 36.9 | 39.7
> | PVT-tiny+PEG+GAP  |23|38.7|41.8
> |PVT-small| 34|40.4|42.2
> |PVT-small+PEG+GAP |34 |43.0|45.2
> |PVT-Medium| 54|41.9|43.2
> |PVT-Medium+PEG+GAP|54|44.3|46.4
>
>
> **Q2:** I also don't completely buy the argument that CPEs allow transformers to have longer input sequences. The real bottleneck is the quadratic complexity of transformers, which CVPT does not address.
>
> **A2:** The role of CPE is to provide better positional encodings for vision transformers. It can result in better performance than the original absolute positional encodings and makes the models generalize well to other resolutions different from the ones used in training. Indeed, CPE has nothing to do with the quadratic complexity bottleneck of transformers, and we admit that it is also an important and interesting problem of vision transformers. Our CPE can be seamlessly combined with the advances on this topic, resulting in a stronger model overall. For example, PVT is a method that alleviates the quadratic complexity bottleneck, and we have shown that PVT+CPE can have significantly improved performance on various vision tasks.
>
> **Q3:** How do the other patches get absolute location information?
>
> **A3:** If the absolute position of any input token is known, the absolute position of all the other tokens can be implicitly inferred by the mutual relation between input tokens because self-attention has a global receptive and CPE uses the convolutions with overlapped windows (i.e., a 3x3 kernel size with stride 1).
>
> **Q4:** The sentence at the end of page 8 could be clearer.
>
> **A4:** Sure, we've revised it in the latest version. When we train the model from scratch, instead of randomly initializing the PEG, we initialize it with the PEG of a well-trained CPVT model and freeze the PEG during the training. The model is trained for 300 epochs and can also achieve 72.3% top-1 accuracy on the ImageNet validation set. This is very close to the model with the PEG trained together (72.4%). This result is also compared with the model with a randomly initialized and fixed PEG (the only difference is that the PEG is not initialized with that in a well-trained CPVT model), which has an inferior performance of 71.3% as shown in Table 5. This suggests that the learned PEG can extract the position information better than the randomly initialized one.

---

> > ### Comment · Reviewer_tvtM · 2022-11-30
> > **Reply to Authors**
> >
> > Hello authors,
> >
> > Thank you for clearing up my confusion. I now understand how CPEs inject absolute location information to all patches. And with regard to the "quadratic complexity" problem, I now see that I misinterpreted the third paragraph of the introduction to be about this.
> >
> > In light of this, I've raised my score to an 8. I think this is a good paper with a straightforward (if not simple) idea and a plethora of experiments with positive signal.

---

### Official Review · Reviewer_Md5K · 2022-10-25

**Confidence:** 4
**Correctness:** 4
**Technical Novelty And Significance:** 4
**Empirical Novelty And Significance:** 3
**Recommendation:** 8

**Clarity, Quality, Novelty And Reproducibility:**

Based on the discussion above, I find hat the proposed PE scheme has novelty. The paper is well written and easy to follow. Although the method description is quite brief, it is clear as it is based on simple concepts. For the same reason, it should be easy to reproduce the results. Also, a brief code snippet is provided as reference.

**Strength And Weaknesses:**

## Strengths
The proposed conditional positional encoding scheme is able to handle longer sequence than the ones used during training, which is a significant restriction of learnable PE schemes. Moreover, similarly to learnable PE schemes, it offers positional equivariance and absolute positional information. Absolute positional information has been shown to be useful for visual tasks. This is a shortcoming of the relative PE methods. Based on these observations, the proposed method offers an advantage over both learnable and relative PE schemes. The PE is achieved by applying equivariant functions (such as convolutions), called Positional Encoding Generator (PEG), on the reshaped feature tokens.

The experiments cover both the ability of CPE to handle larger input sequences (higher resolution images), while also showing that the resulting encoding correctly captures positional information. A solution based on global average pooling, instead of the cls_token is also discussed.

## Weaknesses
The paper mentions that PEG can have "various forms", however in the experimental evaluation a single PEG structure is considered (single layer 3x3 depth-wise convolution). It would be interesting to discuss some alternatives and, possibly, their performance.

Additionally, a single visual transformer method is considered, DeiT. Although, it is expected that similar improvements would occur for other transformer methods, it would be interesting to include at least one more, e.g. the original ViT, T2T [R1] or XCiT [R2].

Also, some additional positional encoding schemes as [R3] and [R4] could be discussed in the related work. Although they mainly address NLP problems, they also offer favourable properties for long sequences.

[R1] Yuan et al., (2021). Tokens-to-Token ViT: Training vision transformers from scratch on imagenet. ICCV

[R2] Ali et al., (2021). XCiT: Cross-covariance image transformers. NeurIPS

[R3] Wang et al., (2020). Encoding word order in complex embeddings. ICLR

[R4] Su et al., (2021). Roformer: Enhanced transformer with rotary position embedding. arXiv

**Summary Of The Paper:**

The paper presents a method for endowing positional information to vision transformer architectures. The proposed conditional positional encoding (CPE) offers translation-equivariance, and it is able to handle longer sequences than the ones encountered during training. In addition, it also captures absolute position, addressing shortcomings of relative PE.

**Summary Of The Review:**

Based on the discussion above, I propose acceptance as the method proposed is novel and offers significant improvements to the classification performance of transformers. Nevertheless, there are some weaknesses that if they were properly addressed, the paper would become much stronger.

## Comments after the rebuttal
I enjoyed the very constructive discussion that took place during the rebuttal. I think that the paper has become much stronger from the technical point of view, and I continue to support acceptance of this work. Regarding the comments about novelty, I agree that some contributions stem from prior work (e.g. Islam et al., 2020), but I still find the focus of this work somewhat different, and thus useful to the community.

Besides the addition of the new results, I agree with other reviewers that the text also needs to be revised based on the discussions that took place. From my side, I would recommend the following:
1) focus more on the ability of the method to generalize to resolutions not seen during training,
2) elaborate on the requirements stated in the beginning of Section 3.2, based on the discussion that took place,
3) include some comments about comparison of the proposed method with other encoding schemes in Section 2,
4) revise this sentence "... and F can be of various forms such as a various types of convolutions and many others", since all PEG forms considered are based on convolutions (e.g. remove "and many others" or add details about specific forms and/or their properties)

---

> ### Author Response · Authors · 2022-11-12
> **PEG benefits other well-known vision transformers like PVT and Swin**
>
> **Q1:** Only a single PEG structure is considered.
>
> **A1:** Other forms of PEG can be that with more depth-wise convolutions, larger kernels and normal (dense) 2D convolutions. We add these experiments to Table~13 in the updated version (B.7 of the appendix). All these designs have similar performance. Therefore, we choose the simplest one.
>
> **Q2:** A single visual transformer method is considered.
>
> **A2:** In Table 12, we also apply our PEG to PVT (Wang et al., 2021) for object detection and semantic segmentation tasks. Our PEG shows consistent and significant performance improvements.
> Besides, in Figure 5 (section B.5), we apply our PEG to Swin. It can boost Swin-tiny from 81.10% to 82.25% top-1 accuracy on the ImageNet dataset.
>
> Table r1. PVT Framework Segmentation on AED20k.
> | Models    | ImageNet Top-1(%) |  Params(M)      |  mIoU（%）|
> |--------|:---------:|------:|----:|
> | ResNet-18 |  69.8| 16 |32.9
> | PVT-tiny|    75.0 | 17|35.7
> | PVT-tiny+PEG  | 77.3 | 17 | 38.0
> | PVT-tiny+GAP|   75.9 | 17 | 36.0
> | PVT-tiny+PEG+GAP  |78.1| 17|38.8
> |PVT-small| 79.8 |28|39.8
> |PVT-small+PEG+GAP | 81.2| 28|44.3
> |PVT-Medium| 81.2 |48|41.6
> PVT-Medium+PEG+GAP|82.7| 48|44.9
>
> Table r2. PVT Framework Object detection on COCO
>
> | Models    |   Params(M)      |  1x mAP（%）| 3x mAP (%)|
> |--------|:---------:|------:|----:|
> | PVT-tiny|   23|36.7|39.4
> | PVT-tiny+PEG  |23|38.0| 41.8
> | PVT-tiny+GAP|  23| 36.9 | 39.7
> | PVT-tiny+PEG+GAP  |23|38.7|41.8
> |PVT-small| 34|40.4|42.2
> |PVT-small+PEG+GAP |34 |43.0|45.2
> |PVT-Medium| 54|41.9|43.2
> PVT-Medium+PEG+GAP|54|44.3|46.4
>
>
> **Q3:** Discussions about other forms of positional encodings in NLP.
>
> **A3:** Thank you for pointing it out. We have included in the revision.

---

### Official Review · Reviewer_eRhn · 2022-10-25

**Confidence:** 4
**Correctness:** 3
**Technical Novelty And Significance:** 3
**Empirical Novelty And Significance:** 3
**Recommendation:** 6

**Clarity, Quality, Novelty And Reproducibility:**

**Clarity & Quality**
Paper is well written, the idea of the paper presented well. There is some conceptual disagreement with some statement (see below) but apart from that everything is clear. There are a bunch of experiments and ablations are done to show effectiveness of proposed method and investigation its attention maps and where we need to place it as well as generalization to other resolutions at test time.

**Novelty**
I think the work lacks of some novelty. As discussed above there are a bunch of works which use convolution as a relative positional embedding in the past. One of the interesting findings is generalization ability which probably was not considered for convolutional augmented transformer models / positional embedding before. Statement about translation invariance I think is weak as we anyway break it by zero-padding.

**Reproducibility**
- Did authors reimplement DeiT or take its offical code and integrated PEG?
- Could authors confirm that the only thing varied in the experiments is positional embedding while training pipeline and other parameters are kept the same?
- A bunch of details are given in the main text and in the Appendix.

**Some other general comments**
- "but it degrades the performance without fine-tuning as later shown in our experiments": there is a bunch of works which showed the same trend, e.g. [7]
- sec 3.1 discussion of no pos.emb. : I think it is not "dramatically degradation" of performance. It is very competitive. Also could authors provide no pos.emb. results in Table 2 to know how it is comparable in generalization regime as it should generalize for all resolutions well?
- typo "in A" -> "in Appendix A"
- In Table 1 is 2D RPE actual 2D or it is stacking of x and y coordinates?
- Sec 3.2: points (1) and (3) contradict each other.
- "CVPT_GAP is uttermost translation-invariant" -> but we have zero padding which breaks translation invariance.
- Sec 4.2 - what is happening with lower resolution? do authors have generalization to lower resolution too?
- There are a bunch of papers proposed not to use CLS token or do some other variant where/how to add it so it is not novel.
- why translation-invariance should be beneficial for ImageNet particularly? We have only centered objects there, maybe it doesn't matter much for this particular dataset.
- Table 2: it is better to have it as graph and put the table into Appendix. Many numbers are hard to parse together.
- Table 3: do authors have results for DeiT-S with GAP?
- typo: "extra number of parameter introduce" -> "extra number of parameter introduced"
- why in Figure 3 why right plot has shifted diagonal by >10 positions? is it effect of PEG module?
- Figure 4: is 384 train resolution or test one?
- did authors have results for base and small architectures to probe generalization to different resolutions?
- Main text, Append A3: I do not fully agree with statement that absolute positional embedding is not translation equivariant. E.g. sin.pos. embedding is designed in the way that if we learn unitary matrix then we can query neighbors, so we can emulate the rel.pos. embedding in the end. Also pushing on the translation equivariance property a lot and then in the propose method which is also not translation equivariant due to the padding and mentioning that actually abs.pos. embedding is needed too for good performance is very confusing.
- Table 10: could authors clarify how they perform ablation without padding?
- Speed of the proposed embedding is investigated but in the end it is better to insert it in several layers - what is then the slowdown?
- Table 12: can we have benchmark without GAP part too?
- Appendix D1: I think it should be reformulated in the way that ImageNet contains mainly centered one object and that is why even no pos.emb. works very well (bag of patches). Also it was shown that patches themselves give huge inductive bias and for particular ImageNet we probably do not need relative position information, so abs.pos. works well too. If we look at the 2d relpos in lambda networks it performs on pair with no pos.emb. which support above statement. So more careful formulation and reasoning is needed in the text.


**Details Of Ethics Concerns:**

No concerns, the paper is about fundamental component of transformer architecture.

Update after rebuttal period: I am raising the score from 5 to 6 and strongly support acceptance of the work though some changes in the text are needed for the final revision (see discussion below).

**Strength And Weaknesses:**

**Strength**
- Demonstrating that simple conv-based relative positional embedding (PEG) performs well for ViT models on ImageNet and improves over the baselines.
- Experiments which demonstrate that proposed embedding provides generalization to other resolutions
- Ablations on the positioning of PEG and number of occurrences.

**Weaknesses**
- This paper applying convolution to model relative positional embedding in the context of ViT. But this was done in the prior works, see [1-5], in many domains, especially in speech recognition. The only novelty is the proper applying conv to the image patch sequence: we reshape back to image to apply conv and then return back to patch sequence. However this is very natural way to apply I think. What is then the main difference with all these [1-5] prior works? Especially in the context that it is better to apply PEG in several layers.
- With current formulations of abs.pos. (or rel.pos.) we can simply use MAE SSL while for the proposed method it is non-trivial as changes the encoding based on the input. A bunch of design choices of positional embeddings remain the main vanilla transformer untouched so many interesting SSL can be applied like MAE or position prediction (see Zhai, S., Jaitly, N., Ramapuram, J., Busbridge, D., Likhomanenko, T., Cheng, J.Y., Talbott, W., Huang, C., Goh, H. and Susskind, J.M., 2022, June. Position Prediction as an Effective Pretraining Strategy. In International Conference on Machine Learning (pp. 26010-26027). PMLR).
- How it is comparable with other baselines with rel.pos. like Convit [6] or variation of abs.pos. e.g. sin.pos. and CAPE [7]?


Prior works and some other relevant works to be cited in the paper:
- [1] Gulati, A., Qin, J., Chiu, C.C., Parmar, N., Zhang, Y., Yu, J., Han, W., Wang, S., Zhang, Z., Wu, Y. and Pang, R., 2020. Conformer: Convolution-augmented Transformer for Speech Recognition. Interspeech 2020.
- [2] B. Yang, L. Wang, D. Wong, L. S. Chao, and Z. Tu, “Convolutional self-attention networks,” arXiv preprint arXiv:1904.03107, 2019.
- [3] A. W. Yu, D. Dohan, M.-T. Luong, R. Zhao, K. Chen, M. Norouzi, and Q. V. Le, “Qanet: Combining local convolution with global self-attention for reading comprehension,” arXiv preprint arXiv:1804.09541, 2018.
- [4] A. Mohamed, D. Okhonko, and L. Zettlemoyer. Transformers with convolutional context for ASR. arXiv, abs/1904.11660, 2019.
- [5] Baevski, A., Zhou, Y., Mohamed, A. and Auli, M., 2020. wav2vec 2.0: A framework for self-supervised learning of speech representations. Advances in Neural Information Processing Systems, 33, pp.12449-12460.
- [6] d’Ascoli, S., Touvron, H., Leavitt, M.L., Morcos, A.S., Biroli, G. and Sagun, L., 2021, July. Convit: Improving vision transformers with soft convolutional inductive biases. In International Conference on Machine Learning (pp. 2286-2296). PMLR.
- [7] Likhomanenko, T., Xu, Q., Synnaeve, G., Collobert, R. and Rogozhnikov, A., 2021. CAPE: Encoding relative positions with continuous augmented positional embeddings. Advances in Neural Information Processing Systems, 34, pp.16079-16092.
- [8] Li, Y., Si, S., Li, G., Hsieh, C.J. and Bengio, S., 2021. Learnable fourier features for multi-dimensional spatial positional encoding. Advances in Neural Information Processing Systems, 34, pp.15816-15829
- [9] KERPLE: Kernelized Relative Positional Embedding for Length Extrapolation https://arxiv.org/abs/2205.09921 NeurIPS 2022
- [10] (AliBi) Ofir Press, Noah Smith, and Mike Lewis. Train short, test long: Attention with linear biases enables input length extrapolation. In International Conference on Learning Representations, 2022.
- [11] (SHAPE) Kiyono, S., Kobayashi, S., Suzuki, J. and Inui, K., 2021, November. SHAPE: Shifted Absolute Position Embedding for Transformers. In Proceedings of the 2021 Conference on Empirical Methods in Natural Language Processing (pp. 3309-3321).
- [12] Dai, Z., Yang, Z., Yang, Y., Carbonell, J.G., Le, Q. and Salakhutdinov, R., 2019, July. Transformer-XL: Attentive Language Models beyond a Fixed-Length Context. In Proceedings of the 57th Annual Meeting of the Association for Computational Linguistics (pp. 2978-2988).
- [13] Su, J., Lu, Y., Pan, S., Wen, B. and Liu, Y., 2021. Roformer: Enhanced transformer with rotary position embedding. arXiv preprint arXiv:2104.09864.


**Summary Of The Paper:**

Positional embedding is an important component in transformers. Moreover, generalization ability to longer sequences with proper positional embedding is also hot topic. Current paper proposes a new positional embedding conditioned on the input (PEG) and not only position itself. This is done vie convolutional layer and tested for ViT on ImageNet data. To preserve structure of the initial image sequence of patches is transformed back to image where conv layer is applied and then output is transformed back to patches sequence. Empirically it is shown that PEG has better generalization ability to other image resolutions and improve results over the baseline positional embeddings.

**Summary Of The Review:**

The paper proposes to use conv layer as a relative positional embedding, so that positional embedding depends on the input itself not its position (however zero padding also introduces dependence on the position). The work is interesting overall with some results on improved generalization to other resolutions at test time, improved results overall compared to some baselines. Ablations are also somewhat interesting. The main concern I have is the novelty. Convolution pos. embedding was proposed in many prior works so it is not clear what else this paper brings. Also comparison is done with not so many baselines and conViT for example (rel.pos. in self attention directly) performs better than results from the paper.

---

> ### Author Response · Authors · 2022-11-12
> **Part 1**
>
> **Q1:** Difference with [1-5].
>
> **A1:** All these works are for other tasks (e.g, speech recognition or machine translation) rather than computer vision. Also, they are mainly for 1D signals and our CPE is for 2D vision images. In contrast, we show that CPE can work in a much wider range of tasks in 2D vision including classification, detection, and segmentation.
>
> **Q2:** CPE for MAE SSL.
>
> **A2:** We admit that it is non-trivial to incorporate CPE with MAE for SSL. But MAE is not the only choice for SSL pre-training. For example, SimMIM can be used with our CPE and it can achieve similar performance to MAE as shown in their paper. Moreover, not all use cases of transformer models need SSL pre-training, and there are numerous applications where SSL is not applicable due to limited data or computational resources. In addition, this issue only arises during the MAE pre-training stage. Nevertheless, if we anyhow want to use MAE, one workaround is that we could use the original sinusoidal positional encodings in the MAE pre-training stage, and then switch to our CPE during the fine-tuning stage (ignoring the mismatched keys).
>
> **Q3:** Comparison with absolute encodings, ConViT [6], and CAPE [7].
>
> **A3:**  **(1)** We compare with absolute sin-cos positional encodings in Table 7 (Section 5.3). Sin-cos positional encodings achieve similar results to the baseline of learnable positional encodings that we compared (72.3% vs 72.2%), and our method still performs better as shown in Table 7. **(2)** First, we would like to highlight that our work is contemporary with these two (see A2 to Reviewer eRhn). ConViT and CAPE were made public in March and June 2021 respectively. Moreover, we have different motivations from these two. ConViT proposes an operation GPSA that can degrade into a convolution if the flexibility of the self-attention operation is not needed. Note CPVT-S-GAP has comparable performance to ConViT with fewer parameters and FLOPS (see Table r0 in the answer to Reviewer oX4R). CAPE chooses to augment the original positional encodings with continuous positions to improve the generalization of the model. Instead, we propose a plug-and-play module that is able to totally replace the original scheme of absolute/relative positional encodings. In addition, as shown in Table 1 in the CAPE paper, directly using CAPE instead of the original positional encodings may impair the performance (from 81.32% to 81.01%) while our method can always result in better performance than the original positional encodings.
>
> **Q4:** Missing references.
>
> **A4:** Thanks. We’ve cited some notable works of these in the revision.
>
> **Q5:** Reproducibility.
>
> **A5:**  **(1)** The official code of DeiT is used, and we integrated PEG into it.  **(2)** Yes, unless specified, only the positional embeddings are replaced and other settings are kept identical.
>
> **Q6:** Performance of no positional embeddings.
>
> **A6:** On ImageNet, the performance drop from 72.2% to 68.2%, considering that only one single change is made (i.e., removing the positional embeddings), is considered significant.
>
> **Q7:** Provide no PE results in Table 2 to know how it is comparable in the generalization regime.
>
> **A7:**  The result is updated in Table 2.  DeiT-tiny (no pos) has 68.2%(224), 68.6%(384),68.4%(448) and 65.0%(512). This performs much worse than our method which has 72.4%(224), 73.2%(384), 71.8%(448) and 70.3%(512).
>
> **Q8:** 2D RPE implementation in Table 1.
>
> **A8:** It’s from the 2D extension from famous relative positional encoding (Shaw et al., 2018),  which is implemented by the tensor2tensor repo.
>
> **Q9:** On the translation invariance of CPE.
>
> **A9:** Yes, zeros paddings can break the translation invariance (which should be translation equivalence, more accurate), making the model not strictly translation-invariant. But compared to the original absolute positional encodings, our CPE can keep the translation invariance as much as possible. We admit that if we consider the strict translation invariance, our model is not translation-invariant, but neither is the standard convolution with zero paddings. In fact, it is impossible to have a model with both absolute positional information and translation invariance. But as discussed in our paper, both are important to performance. Thus, a good model should achieve the best trade-off between them. Here, we argue that our method achieves a better point than the previous absolute fixed/learnable positional encodings.
>
> **Q10:** Sec 4.2 - what is happening with lower resolution? Generalization to lower resolution too?
>
> **A10:**  We test the results on the lower resolution 160 as you suggested and update the results in Table 2. Although the overall trend is that all models‘ performance drops on a lower resolution, our method can attain better accuracy than others across all scales (tiny, small, and base). For example, CPVT-Tiny outperforms DeiT by 1.2% in accuracy.

---

> ### Author Response · Authors · 2022-11-12
> **Part 2**
>
> **Q11:** There are a bunch of papers proposed not to use CLS token or do some other variant where/how to add it.
>
> **A11:** To the best knowledge, our work is the first one to use GAP instead of CLS token in a vision transformer on the date when it is made public (Feb 2021).
>
> **Q12:** Why translation-invariance should be beneficial for ImageNet particularly? We have only centered objects there, maybe it doesn't matter much for this particular dataset.
>
> **A12:** The objects on ImageNet are not necessarily centered in the images. Therefore, the translation invariance is helpful on ImageNet. Moreover, there might be multiple objects in a single image. In this case, the category of the object closest to the image center would be the target category of the image.
>
> **Q13:** Table 2 is better as a graph.
>
> **A13:** Sure.
>
> **Q14:** Table 3: do authors have results for DeiT-S with GAP?
>
> **A14:** DeiT-S with GAP obtains 80.2% top-1 accuracy, updated in Table 3.
>
> **Q15:** Typo in "introduce".
>
> **A15:** Fixed.
>
> **Q16:** In Figure 3 why it has a shifted diagonal?
>
> **A16:** Not always. The updated Figure 6 (appendix) shows the other two heads within the same layer are diagonal.
>
> **Q17:** Fig. 4: is 384 train resolution or test one?
>
> **A17:** Test. All models are trained with 224.
>
> **Q18:** Base/small models to probe generalization to other resolutions?
>
> **A18:** CPVT-B/S are given in Table 2.
>
> **Q19:** I do not fully agree that absolute PE is not translation equivariant.
>
> **A19:** Yes, in theory, it is possible that the model can learn to transform the absolute positional encodings into the relative ones, thus being translation equivariant. While it is possible, there is no explicit incentive to do so. CPE instead makes translation equivariance as one of its intrinsic properties. In addition, whether zeros paddings break the translation equivariance is discussed in A9 to Q9.
>
> **Q20:** How to perform ablation w/o paddings?
>
> **A20:** We simply removed the paddings in our implementation.
>
> **Q21:** Speed slowdown if inserted in several layers?
>
> **A20:** CPE is very lightweight and barely slows down the model. Table 4 shows that the throughputs of our model (CPE in several layers) and DeiT are very close (2536 vs 2500 imgs/s).
>
> **Q22:** Benchmark without GAP?
>
> **A22:** We run the experiment without GAP and report the result in Table12 in the revision.  PVT-tiny with PEG obtains 77.3% top-1 accuracy on ImageNet validation set, 38.0% mIoU on ADE20K, and 38.0%mAP(1x), 41.8mAP(3x) on COCO, which outperforms PVT-tiny with GAP by a clear margin. This suggests that PEG is critical for good performance. PVT-tiny+GAP+PEG performs best.
>
> **Q23:**  More formulation and reasoning in D1.
>
> **A23:** It is hard to say that the model with no PE works well. It only attains 68.2%. As mentioned in A12, the objects on ImageNet are not necessarily centered in an image. If there are multiple objects with different categories in an image, the category of the object closest to the image center is the category of the image. This is why absolute position information is still important. In addition, the lambda networks use relative positional encodings only thus and we partly attribute the inferior performance to the lack of absolute information. As mentioned, the absolute position information is also important for the ImageNet classification task. Finally, the results of lambda networks and DeiT are not directly comparable due to many other differences between the two works.

---

> ### Author Response · Authors · 2022-11-15
> **Have we cleared your doubts?**
>
> Hello,
>
> We would like to know if these answers well address your concern so that we can further respond. Any new questions are welcomed, please let us know.
>
> Regards,
> The authors

---

> > ### Comment · Reviewer_eRhn · 2022-11-18
> > **Response to authors' comments**
> >
> > Dear authors,
> >
> > Thanks for detailed responses and additional ablations and evaluations! Please find below some of my comments:
> >
> > > To the best knowledge, our work is the first one to use GAP instead of CLS token in a vision transformer on the date when it is made public (Feb 2021).
> >
> > Indeed, one of the references I had in mind is https://arxiv.org/pdf/2105.02723.pdf (I remember it was one of the first works I saw on using pooling instead of cls token). Do you then claim novelty by removing cls token? Sadly, I am not sure if it is valuable to claim now as novelty after 2 years when you first made the work to be public.
> >
> > > The objects on ImageNet are not necessarily centered in the images. Therefore, the translation invariance is helpful on ImageNet. Moreover, there might be multiple objects in a single image. In this case, the category of the object closest to the image center would be the target category of the image.
> >
> > So far it is speculation seems to me. And results with no positional embedding which are surprisingly very high prove opposite, that relative information and even position itself is not so important. Do you have clear empirical justification? Especially in the context that you use padding to break translation equivariance (so with no padding it is working worse - maybe we need some disentangled experiments to quantify). Correct me, if I misunderstand.
> >
> > > Not always. The updated Figure 6 (appendix) shows the other two heads within the same layer are diagonal.
> >
> > But why shift in diagonal can happen? Any idea?
> >
> > > On absolute PE is not translation equivariant.
> >
> > Yep, I got your point on the stronger explicit bias you introduce. Then I am ok if the tone of text is changed saying about "stronger explicit bias", rather than saying that absolute PE is not translation equivariant.
> >
> > > We simply removed the paddings in our implementation.
> >
> > Does it mean that we will have smaller number of tokens later (as output has a bit smaller size as an image)?
> >
> > > A23
> >
> > If you look e.g. at CAPE work for speech domain you will see huge drop for no positional embedding :) in that sense drop of several % of accuracy is not drastic as reported e.g. in speech in prior work, which is also reasonable. But I got your point. I am ok with your thoughts but I still think that tone should be a bit change in the paper as I believe we need in future some more disentangled experiments and analysis to understand better how absolute and relative positions works specifically for ImageNet and what are the main factors.
> >
> > Best regards.

---

> > > ### Author Response · Authors · 2022-11-18
> > > **No positional embedding also produces drastic drop in vision**
> > >
> > > Thanks for the further comments.
> > >
> > > **Q1:**  Indeed, one of the references I had in mind is https://arxiv.org/pdf/2105.02723.pdf (I remember it was one of the first works I saw on using pooling instead of cls token). Do you then claim novelty by removing cls token?
> > >
> > > **A1:** It is unfortunate to claim it now but our originality cannot be erased.
> > >
> > > The work you mentioned  (https://arxiv.org/pdf/2105.02723.pdf) however is still using the class token (we’ve carefully checked this paper and their code). Maybe you are referring to another paper?
> > >
> > > **Q2:** Results with no positional embedding which are surprisingly very high prove the opposite, that relative information and even position itself are not so important. Do you have a clear empirical justification? Maybe we need some disentangled experiments to quantify.
> > >
> > > **A2:** Speech recognition is a *sequence-to-sequence prediction task* where position information plays a more important role (note position is still kept in the results). Instead, we work on a *sequence-to-scalar* task where only a classification label is in the result. Therefore, it is very likely that no PE damages speech recognition so much but not too obvious in vision. The top-1 accuracy gap between DeiT/CPVT and DeiT-no position on ImageNet is **more than 4%**, which is regarded as **very significant** in the vision community.
> > >
> > > From our perspective, it is a reasonable speculation. It would be better, but impractical to make disentangled experiments to quantify this issue because ImageNet doesn’t provide such annotations, where only a single label is provided. A disentangled experiment would require much finer annotations, e.g., bounding boxes for these objects.
> > >
> > > **Q3:** The updated Figure 6 (appendix) shows the other two heads within the same layer are diagonal. But why the shift in diagonal can happen? Any idea?
> > >
> > > **A3:** As stated in Figure 3, the plotting is based on the second attention block. If we plot the attention score map of the third block for DeiT in Figure 6 (updated again), we also observe the off-diagonal phenomenon. This means even DeiT doesn’t always have diagonal scores. Therefore, it seems non-deterministic whether it is diagonal or not. Nevertheless, it’s important that both DeiT and our method can learn local biases or relations. The specific form (diagonal or off-diagonal) is likely to be agnostic and may not be fixed.
> > >
> > > **Q4:** The tone of text should be changed to "stronger explicit bias".
> > >
> > > **A4:**  Thanks and we have rephrased it in the latest revision.
> > >
> > > **Q5:** Does it mean that we will have a smaller number of tokens later (as output has a bit smaller size as an image)?
> > >
> > > **A5:**  Yes.  The number of tokens is 12x12 instead of 14x14 for an image of 224x224.
> > >
> > > **Q6:** Huge drop for no positional embedding in CAPE.
> > >
> > > **A6:** Please refer to A2.
> > >
> > > Thanks again. We are looking forward to hearing from you for further feedbacks.

---

> > > > ### Author Response · Authors · 2022-11-22
> > > > **Any new questions?**
> > > >
> > > > Dear Reviewer eRhn,
> > > >
> > > > Do our latest replies resolve your question well? We are willing to discuss any further issues.
> > > >
> > > > Regards,
> > > >
> > > > The Authors

---

> > ### Comment · Reviewer_eRhn · 2022-11-23
> > **Discussion**
> >
> > Dear authors,
> >
> > Thanks for all responses and gentle ping! It took a bit of time to go over the last revision. I like now Table 2, smoother discussions and additional ablations and comparisons. It looks stronger to me now. Having in mind your mentioning about first publishing the work in beginning of 2021, I believe additional comparisons with recent works in the Appendix are strong proof of the effectiveness of PEG. But I am still concerned on the fact that you combine absolute and RPE together: I agree with Reviewer oX4R and looking at current results in Table 7 and 10 - RPE 2D 70.5, PEG with no padding 70.5, PEG with padding (=absolute learnable + RPE) 72.4, PEG with padding 0-5 73.4. Could you clarify here:
> >
> > 1) RPE 2d is used in every layer? Do you know what PEG without padding will give if we use in every layer?
> > 2) Could it be that in the end PEG is so good and comparable with recent works (Appendix E2) because you use combination of absolute and RPE while other do not use this?
> >
> > Small comments (do not affect final decision):
> > - "Similar to the original positional encodings in DeiT, the conditional positional encodings are also added to the input sequence, as shown in Figure 1 (b)" - could you clarify if you add PEG after first transformer block or add it directly on the input patches?
> > - typo, page 5: "before the final classification layer of CPVT. CPVT " -> put comma instead of dot.
> >
> > Happy to discuss more,
> >
> > Reviewer eRhn.

---

> > > ### Author Response · Authors · 2022-11-23
> > > **PEG and RPE**
> > >
> > > Dear Reviewer eRhn,
> > >
> > > We appreciate your responsiveness and thoroughness very much, it is so crucial to us. Here are the answers to the new questions, please find yourself comfortable to advance the discussion.
> > >
> > > **Q1:** Is 2D RPE used in every layer? Do you know what PEG without padding will give if we use it in every layer?
> > >
> > > **A1:** No. In Table 7, 2D RPE is used only once. Both LE (learnable absolution encoding from DeiT) and 2D sin-cos in the first two rows also occur once. They are made mostly comparable.
> > >
> > > PEG without padding in every encoder shouldn’t have good performance. Particularly,  DeiT has 12 layers, if we don’t use padding, the number of tokens will decrease by depths, e.g. it starts from 14x14 in the first block and is already reduced to 2x2 in the sixth one. Therefore, it can hardly outperform others when its power is diminishing yet it is already inferior in the beginning.
> > >
> > > **Q2:** Could it be that PEG is so good and comparable with recent works (Appendix E2) because you use a combination of absolute and RPE while others do not use this?
> > >
> > > **A2:** Yes, we suggest that the better performance of PEG may come from simultaneously providing the absolute positions (from zero paddings) and the local relationships between tokens (i.e., like RPE as you mentioned).
> > >
> > > However, this does not imply that our PEG can be simply viewed as a combination of the absolute position encodings and RPE. It turns out that our PEG means more than just a combination. For example, it can seamlessly cope with input images of various sizes (as discussed in Table 2 and 12). In contrast, the original sin-cos or learnable absolute positional encoding have trouble dealing with it. Moreover, our PEG blends the models with more translation equivariance as discussed before. This is also why we termed it “conditioned” since our positional encodings are dynamically generated according to the nearby tokens.
> > >
> > > **Q3:** Could you clarify if you add PEG after the first transformer block or add it directly on the input patches?
> > >
> > > **A3:** After the first transformer encoder block.
> > >
> > > **Q4:** The dot typo on Page 5.
> > >
> > > **A4:** Thanks. It will be fixed.

---

> > > ### Author Response · Authors · 2022-11-24
> > > **Updates on PEG w/o Padding for every block.**
> > >
> > > To complete our analysis, we’ve carried out this experiment on PEG w/o padding at every layer. Specifically, DeiT has 12 transformer encoder blocks in total. Since the resolution of the sixth block is already reduced to  2x2 (i.e. up to 5 tokens including the class token), we have to use 2x2 for the remaining 6 blocks. Due to the much-decreased tokens (reduced FLOPs too), this no-padding version behaves poorly, as shown in Table r1.  In fact, this setting is quite strange in vision because down-sampling is applied for every block.
> > >
> > >
> > > **Table r1. PEG w/o padding for every block in DeiT**
> > >
> > > | Models    |  FLOPS   | ImageNet Top-1(%) |
> > > |--------|:---------:|------:|
> > > | DeiT-tiny |1.3G |72.2 |
> > > | DeiT- PEG w/o pading for every block| 0.3G | 59.7 |

---

> > > > ### Comment · Reviewer_eRhn · 2022-11-25
> > > > **Discussion**
> > > >
> > > > Thanks for answers and conducting extra analysis.
> > > >
> > > > I think I converged to the main point which I don't understand and still don't buy from current experiments. It is confirmation that your combination of absolute and relative positions via PEG is the best way compared to simple combination of known RPE and APE (absolute one).
> > > > I asked about 2d RPE to understand if it is fair comparison with PEG (no padding) used in one place or not. Right now it is truly comparable as you use only once both of them (as far as I know RPE is used in every transformer block not only in first layer in other domains like NLP and speech), so then I can conclude that RPE either 2d or via PEG (no padding) gives the same results:  Table 7 and 10 - RPE 2D 70.5, PEG with no padding 70.5.
> > > >
> > > > Reasonable justification of PEG (with padding) for me will be the following experiments:
> > > > - (a) Use RPE 2d and sin/cos or LE absolute at the beginning: x + APE -> transformer block with RPE -> the rest of the model
> > > > - (b) Use LE / sin-cos not only at the beginning but also in next 5 layers to have similar thing as 0-5 PEG configuration.
> > > >
> > > > (b) will show that with providing absolute information at every of 0-5 layers is good but not enough as we use relative information with PEG at the same time.
> > > > (a) will show that PEG is not just a combination of APE and RPE but it is doing something more, or more efficiently, or with proper inductive bias (your equivariance explanation).
> > > >
> > > > I agree that PEG provides better generalization to other resolutions than APE, but it could be that (a) option can deal with multi resolutions too as there is RPE to fix bad effects. However I still do believe that PEG will be better anyway than (a) for multi resolutions. But the core of the paper is not just better generalization, that is why I still think justification as (a) is needed.
> > > >
> > > > One side questions: how do you exactly do implementation of 2d RPE? is it stack of RPE for x and RPE for y coordinates?
> > > >
> > > > Thanks!

---

> > > > > ### Author Response · Authors · 2022-11-27
> > > > > **Comparison with APE+RPE, and LE at Pos 0-5**
> > > > >
> > > > > Dear Reviewer eRhn,
> > > > >
> > > > > Many thanks for the prompt reaction and the experiment instructions. We address the new questions as follows.
> > > > >
> > > > > **Q1:**  (a) Use RPE 2d and LE absolute at the beginning: x + APE -> transformer block with RPE -> the rest of the model.
> > > > >
> > > > > **A1:** This setting achieves 72.4% top-1 accuracy on ImageNet, which is comparable to a single PEG (72.4%).  Nevertheless, this experiment doesn’t necessarily indicate that our CPE is a simple combination of APE and RPE. When tested on different resolutions, setting (a) cannot scale well compared to ours (Table r1). RPE is not able to adequately mitigate the performance degradation on top of LE. This shall be seen as a major difference.
> > > > >
> > > > > *Table r1. Comparison with LE+RPE at various resolutions*
> > > > >
> > > > > | Models    |  Positional Params   | Top-1(%) 224 | Top-1(%) 160 |Top-1(%) 384 |Top-1(%) 448|Top-1(%) 512|
> > > > > |--------|:---------:|------:|------:|------:|------:|------:|
> > > > > | DeiT-tiny (LE+RPE, setting a) |40011|72.4 | 65.6| 70.8|68.4| 65.6|
> > > > > | DeiT-tiny (PEG at Pos 0) |1920|72.4 |66.8|73.2|71.8|70.3|
> > > > >
> > > > >
> > > > > **Q2:**  (b) Use LE / sin-cos not only at the beginning but also in the next 5 layers to have a similar thing as 0-5 PEG configuration
> > > > >
> > > > > **A2:** This setting achieves 72.7% top-1 accuracy on ImageNet, which is 0.7% lower than PEG (0-5).  This setting verifies that it is beneficial to have more of LEs, but not as good as ours. It is as expected since we exploit relative information via PEGs at the same time.
> > > > >
> > > > > *Table r2. Comparison of various PEs at more positions in DeiT*
> > > > >
> > > > > | Models    |  FLOPS   | ImageNet Top-1(%) |
> > > > > |--------|:---------:|------:|
> > > > > | DeiT-tiny (LE 0) |1.3G |72.2 |
> > > > > | DeiT-tiny (PEG 0) |1.3G |72.4 |
> > > > > | DeiT-tiny (LE 0-5) |1.3G |72.7 |
> > > > > | DeiT-tiny (PEG 0-5)| 1.3G | 73.4 |
> > > > >
> > > > > **Q3:**  How do you exactly do the implementation of 2d RPE? Is it a stack of RPE for x and RPE for y coordinate?
> > > > >
> > > > > **A3:** Yes, we stack RPE for x and y, like the RPE in the Swin transformer. We exhibit the snippet here,
> > > > > ```python
> > > > > self.relative_position_bias_table = nn.Parameter(torch.zeros((2 *H - 1) * (2 * W - 1), num_heads))
> > > > > coords_h = torch.arange(H)
> > > > > coords_w = torch.arange(W)
> > > > > coords = torch.stack(torch.meshgrid([coords_h, coords_w]))
> > > > > coords_flatten = torch.flatten(coords, 1)
> > > > > relative_coords = coords_flatten[:, :, None] - coords_flatten[:, None, :]
> > > > > relative_coords = relative_coords.permute(1, 2, 0).contiguous()
> > > > > relative_coords[:, :, 0] += H - 1
> > > > > relative_coords[:, :, 1] += W - 1
> > > > > relative_coords[:, :, 0] *= 2 *W - 1
> > > > > ```
> > > > >
> > > > > **Summary**
> > > > >
> > > > > To conclude, we prove our method is distinct from the existing positional encoding schemes (including their combinations)  in the following aspects,
> > > > >
> > > > > - Strong generalization performance at variable input resolutions
> > > > > - Much more efficient to imbue the positional information
> > > > > - Straightforward to implement and hassle-free for deployment
> > > > >
> > > > > We have to emphasize why it is so important to have good generalizations for dynamic resolutions. It may seem just a nice feature for classification, but it creates a *fundamental impact for downstream tasks* like segmentation and detection (see Table 12), where the performance boost can be prominent. This feature shall be viewed as one of the key differences to other positional encoding approaches of Vision Transformers.
> > > > >
> > > > > By simultaneously *offering both absolute and relative positional information*, PEG substantially enhances the performance of a variety of models and tasks. However, it introduces *much fewer parameters* (e..g by **20 times w.r.t. LE+RPE** in Table r1). PEG also enjoys *simplicity in implementation and deployment* being an out-of-box plugin to use in both plain and pyramid ViTs, while RPEs are non-trivial to incorporate as it involves too much intrusion into the self-attention design.
> > > > >
> > > > > We hope to relay the discussion until convergence.
> > > > >
> > > > > Regards,
> > > > >
> > > > > The Authors

---

> > > > > > ### Comment · Reviewer_eRhn · 2022-11-29
> > > > > > **Raising score from 5 to 6**
> > > > > >
> > > > > > Dear Authors,
> > > > > >
> > > > > > Thanks for productive discussion and extensive experiments. I am more confident now with your results and raise my score to 6. For the final revision I recommend to include these latest ablations in the main body but also revisit a bit the text to highlight two positions of the main contributions: 1) explicit almost equivariance inductive bias 2) improved generalization to out of training resolutions. Right now I still think text is needed some work based on the latest ablations to highlight more generalization to other resolutions (and maybe compare with some prior works on that like CAPE, LeViT as this generalization is your selling point) and what exactly PEG is doing and how it is different from simple combination of absolute and relative positional embedding. I would like to raise my score even further but in the light of current text prefer to stick to 6 though I strongly support acceptance of the work now.
> > > > > >
> > > > > > Best,
> > > > > > Reviewer.

---

> > > > > > > ### Author Response · Authors · 2022-11-30
> > > > > > > **Thanks for the support!**
> > > > > > >
> > > > > > > Dear Reviewer eRhn,
> > > > > > >
> > > > > > > Thanks for concluding the discussion and supporting our work. We take pleasure in constructive communication between us.  We will accommodate these ablations into the final revision and spotlight the contributions.
> > > > > > >
> > > > > > > Best regards,
> > > > > > >
> > > > > > > The Authors

---

> > > > > ### Author Response · Authors · 2022-11-29
> > > > > **Update on setting (a) with GAP**
> > > > >
> > > > > Dear Reviewer eRhn,
> > > > >
> > > > > We include one more experiment with GAP under setting (a) to show that PEG benefits more from GAP than LE+RPE, which suggests PEG might be doing more than just a simple combination of ABE and RPE (see Table r3).
> > > > >
> > > > >
> > > > > *Table r3. GAP boosts more for PEG than LE+RPE*
> > > > > | Models    |  Positional Params   | Top-1(%) 224 |
> > > > > |--------|:---------:|------:|
> > > > > | DeiT-tiny (LE+RPE, setting a, w/ GAP) |39819|72.8 |
> > > > > | DeiT-tiny (PEG at Pos 0, w/ GAP) |1920| **73.3** |

---

### Official Review · Reviewer_oX4R · 2022-10-26

**Confidence:** 4
**Correctness:** 4
**Technical Novelty And Significance:** 2
**Empirical Novelty And Significance:** 2
**Recommendation:** 5

**Clarity, Quality, Novelty And Reproducibility:**

- Clarity
The paper is written clearly and it is easy to understand.

-Quality
The quality of the paper is good.

-Novelty
The novelty and technical contribution of the approach is incremental. As mentioned in the weakness, the fact that convolution with zero padding learns implicit positional information is trivial (Islam et al., 2020). Despite the use of the name CPE, the actual implementation uses depth-wise convolution to describe the local pattern. It is considered comparable to CvT except for the location of the convolution.

- Reproducibility
All the details of implementations and evaluations are provided sufficiently to reproduce the reported results.


**Strength And Weaknesses:**

The motivation for work based on the analysis of existing positional encodings is clear, and the paper is easy to understand. Exhaustive experiments are sufficient to show the superiority of the proposed method. The proposed CPE satisfies all the requirements for the desired positional encodings and is easy to combine with standard Transformers. Moreover, it can generalize to the input sequences with arbitrary lengths.

The paper argues that modeling local relationships is sufficient to replace positional encoding. However, the term "conditioned" is ambiguous in the fact that the practical performance gains result from zero paddings. In other words, the suggested CPE depends on zero paddings that serve as an anchor to offer absolute positional information rather than the local relationship. For instance, PEG without zero padding is inferior to PEG with zero padding, implying that the performance gain of CPE is dependent on zero padding. Furthermore, the fact that convolution with zero padding learns implicit positional information is already well-known and trivial (Islam et al., 2020). Various forms other than convolution with zero paddings should be suggested to demonstrate the effectiveness of CPE.
Although Section 5.1 attempted to clarify the suggested CPE as positional encoding, it should be considered as simply combining the original tokens and additional tokens that aggregate local information among pixels using convolution, in the same way as CvT (Wu et al., 2021) uses convolutional projection to aid in local context modeling.
Besides positional encodings, either absolute or relative, several recent attempts to incorporate positional information differently, such as ConViT (d’Ascoli et al., 2021) and CoAtNet (Dai et al., 2021). It is better to add and compare the experimental result with these approaches.

Minor comments
Comparison of different plugin positions at 0 and -1 results are shown in Table 6 left, not Table 6 right. (Section 5.2)


**Summary Of The Paper:**

This paper introduces a new alternative to positional encoding for vision Transformers. They suggest desired properties of positional encoding for visual tasks based on a comparison of existing positional encodings, such as absolute or relative positional encodings. The suggested conditional positional encoding (CPE) satisfies the requirements. Existing vision Transformers can readily incorporate CPE without requiring any implementation changes. As a result, CPE consistently improves the performance of ViT on image classification, segmentation, and object detection tasks. Furthermore, global average pooling replacing a class token boosts the performance further.

**Summary Of The Review:**

The proposal is simple and effective, however does not provide sufficient novelty and technical contribution warranting the paper acceptance.

---

> ### Author Response · Authors · 2022-11-12
> **PEG encodes both absolute and relative positional information**
>
> $\textbf{Q1}$: The suggested CPE depends on zero paddings rather than the local relationship.
>
> $\textbf{A2}$: As mentioned in our paper, zero paddings provide absolute positional information while PEG encodes the local relative relationship between tokens, both of which are of great importance. As shown in Table 5 and Table 10, without zeros paddings, the accuracy drops from 72.4% to 70.5%. However, if PEG is totally removed, the accuracy further drops from 70.5% to 68.2%. This suggests that the local relationship encoded by PEG is crucial as well and zero paddings alone are not sufficient.
>
> $\textbf{Q2}$: Relation to CvT (Wu et al., 2021), ConViT (d’Ascoli et al., 2021) and CoAtNet (Dai et al., 2021). Compare the experimental results with CvT/ConViT/CoAtNet.
>
> $\textbf{A2}$: Yes, our work shares some similarities with these works. But we would like to claim our originality because our work was made public in Feb 2021, and all these works came after this date. Nevertheless, our method is still comparable to or better than these approaches (Table r0 below, also E.2 in the appendix). To make fair comparisons, we categorize them into two groups: **plain** and **pyramid** models. Our plain model CPVT-S-GAP outperforms ConViT-S by 0.2% with 4M fewer parameters and 0.8G fewer FLOPs. To be comparable, we adapt our method onto two popular pyramid frameworks PVT and Swin, which show advantages against CvT and CoAtNet.
>
> Table r0. Comparison with other competent approaches on ImageNet
>
> | Models    | Type |  Params(M)  | FLOPS (G)   | ImageNet Top-1(%) |
> |--------|:---------:|------:|------:|------:|
> | DeiT-small | Plain | 22M | 4.6G |79.9 |
> | ConViT-S | Plain | 27M | 5.4G  | 81.3 |
> | **CPVT-S w/ GAP (Ours)** | Plain  | 23M | 4.6G | **81.5 (+1.4)** |
> | CoAtNet | Pyramid | 25M | 4.2G | 81.6 |
> | CvT-13 | Pyramid | 20M | 4.5G | 81.6 |
> | PVT-Small | Pyramid | 25M |  3.8G | 79.8 |
> | **PVT-Small w/ PEG and GAP** | Pyramid | 25M |  3.8G | 81.2 (+1.4) |
> | Swin-tiny | Pyramid | 29M | 4.5G | 81.3 |
> | **Swin-tiny w/ PEG and GAP** | Pyramid | 29M | 4.5G | **82.3 (+1.0)** |
>
> Note CvT uses a depth-wise convolution in $q$-$k$-$v$ projection which they call it *Convolutional Projection*. Instead of using it in all layers, we put only one of such design into DeiT-tiny and train such a model from scratch under strictly controlled settings. We insert it in the position 0 as in our method. The result is shown in Table r1 below.  This CvT-flavored DeiT achieves 70.6\% top-1 accuracy on ImageNet validation set, which is lower than ours (72.4\%). Note that $q$-$k$-$v$ projections in CvT utilize three depthwise convolutions, therefore, this setting has more parameters than ours. This attests the difference of CvT and CPVT, verifying our advantage by learning better position encodings other than inserting them in all layers to have the ability to capture local context and to remove ambiguity in attention.
>
> Table r1. Comparison with CvT-flavored DeiT on ImageNet validation set
> | Models | Params | ImageNet Top-1(%) |
> |--------|:---------:|------:|
> |CPVT-tiny |  5681320| 72.4 |
> |DeiT+ Convolutional Projection |5685352|70.6|
>
>
> $\textbf{Q3}$: Various forms other than convolution with zero paddings should be suggested to demonstrate the effectiveness of CPE.
>
> $\textbf{A3}$: Please see Table 13 in B.7. According to the results, we use the simplest form (a depthwise 3x3) as our default implementation. It indicates that this design is enough to provide good positional information.
>
> $\textbf{Q4}$: Typos in Section 5.2.
>
> $\textbf{A4}$: Thank you for pointing it out. We have fixed it in the revision.

---

> ### Author Response · Authors · 2022-11-21
> **Looking forward to hearing your response**
>
> Dear Reviewer oX4R,
>
> Many thanks for your valuable remarks.
>
> To make the best use of the discussion period and to improve our work, we'd like to hear your response to know whether we have addressed your concerns or there are any new questions arise.
>
> We hope to resolve your doubts with our best effort. Please share your thoughts on viewing our reply.
>
> Regards,
>
> The authors

---

> ### Author Response · Authors · 2022-11-30
> **Sincerely looking forward to your feedbacks**
>
> Dear Reviewer oX4R,
>
> Thanks again for the reviews.
> We expect any further feedback to enhance our paper and address your rest concerns before the deadline.  We are desirous to have a productive conversation.
>
> Sincerely,
>
> The Authors

---

### Author Response · Authors · 2022-11-12
**General response and revision update notice**

Dear reviewers,

We thank everyone's effort for the precious review. The paper has been moderately revised according to the suggestions received. We've carefully examined all the questions to write the answers. Please feel encouraged to discuss actively with us if any new doubts occur.

Here is a list of updates (by page order) in the paper and appendix:

- Included related works on positional encodings and similar designs in Speech & NLP. (Section 2.2)
- Added lower resolution results and DeiT without position encoding (baseline)  in Table 2. (Section 4.2)
- Revised the sentence at the end of page 8 (suggested by Reviewer tvtM) to make it clearer. (Section 5.1)
- Fixed a typo mentioned by Reviewer  oX4R. (Section 5.2)
- Added small and medium model results in Table 12 based on the  PVT framework. (B.6)
- Added other forms of PEG design in Table 13. (B.7)
- Added extra figures of attention scores in other two heads to show the diagonal shift is not always the case. (E.1)
- Added comparisons with CvT, ConViT, CoAtNet according to Reviewer oX4R. (E.2)

Sincerely,

The Authors

---

### Decision · Program_Chairs · 2023-01-20

**Decision:**

Accept: poster

**Justification For Why Not Higher Score:**

Limited novelty and unclear contributions.

**Justification For Why Not Lower Score:**

Extensive experimentation of convolution as position encoding.

**Metareview: Summary, Strengths And Weaknesses:**

This is a borderline paper. there has been lot of back and forth in discussion among reviewers. On one hand, the paper does a good job in comparing the role of convolutions in encoding positional information. On the other hand, claims about novelty are false and many techniques are already established in earlier works. However the reviewers still believe there is value in the experiments presented in the paper and I agree with them. One of the reviewers strongest objections against paper are based on the novelty of the approach, author's acknowledge that in the response to an extent and this should be corrected in the final version with referencing the right earlier works for the method and limit the paper claims to experimentation and comparisons. I suggest acceptance assuming authors will correct this in final version.

Pros
 - Extensive experimentation and comparison with existing approaches.
- Evaluation of different resolution inputs.

Cons
- Limited novelty.
- Overclaiming contributions/ missing comparisons to some existing works.

I strongly suggest rewriting the final version as per one of the reviewers suggestions - text itself should be rewritten to not claim novelty of the convolution layer with padding as a positional embedding, but rather in-depth analysis of conv block, analysis if it is learning both relpos and abspos, and generalization to other resolutions not used during training.

**Note From Pc:**

if the above contains the word "oral" or "spotlight" please see: "oral" presentation means -> notable-top-5% and "spotlight" means -> notable-top-25%. As stated in our emails, we are disassociating presentation type from AC recommendations

**Summary Of Ac-Reviewer Meeting:**

There has been lot of back and forth during the discussion between reviewers.

Drawback in the submission from the reviewers.
-  The fact that convolution with zero padding learns implicit positional information is already known (Islam et al., 2020), and a convolutional stem is often used for transformer networks; the authors attempt to reblend it under the name of CPE, the actual implementation is depth-wise convolution + zero padding, which is a trivial variant of existing convolutional stems.
The authors' rebuttal does not directly respond to this important issue. While the submission gives good performance with extensive experimental analysis, I don't see a sufficient novelty for a top-tier publication.

Advantages in the submission
- Ultimately I think this paper offers good experiments and insights that other researchers could benefit from. While the paper is simple and marginally novel, I’m not sure this is a good reason to reject the paper, although I may be wrong.
- But I believe that having marginal novelty or even absence of novelty should not be the reason to reject the work in case deep analysis, empirical or theoretical analysis are done even for known things and some helpful in. In this light I believe the text itself should be rewritten to not claim novelty of the convolution layer with padding as a positional embedding, but rather in-depth analysis of conv block, analysis if it is learning both relpos and abspos, and generalization to other resolutions not used during training.